# Autonomous Knowledge Integration Enables Efficient Cross-Modality Prompt Transfer

## Abstract

Cross-Modality Prompt Transfer (CMPT), which transfers prompts pretrained on a data-rich modality (e.g., text), seeks to address initialization instability and data scarcity in prompt tuning of pretrained transformers for tasks with limited data. However, prior research in CMPT typically transfers all source prompt vectors without accounting for vector redundancy or incompatibilities. This indiscriminate transfer may include irrelevant or even detrimental vectors, thereby negatively impacting the performance of the target task. To mitigate this issue, we propose *Selective Cross-Modality Prompt Transfer* (S-CMPT), a method that automatically identifies and transfers only the most relevant prompt vectors. S-CMPT employs a lightweight attention mechanism to select vectors that are most pertinent to the target task. Furthermore, we introduce a simple regularization term to encourage the selection of diverse and non-redundant vectors. The selected vectors are subsequently adapted to the target task through a linear projection layer. As a result, S-CMPT achieves impressive simplicity and effectiveness, demonstrating significant accuracy improvements (+4.42% over prompt tuning and +0.65% over state-of-the-art prompt-based methods) while utilizing far fewer vectors, offering an efficient solution for data-scarce prompt tuning applications.

## 1 Introduction

Pretrained Transformers (Vaswani et al., 2017) have demonstrated remarkable effectiveness across various domains (Liu et al., 2023; Zhao et al., 2023). Although this capability scales with model size (Brown et al., 2020), it introduces significant parameter inefficiency: full finetuning requires storing an entire model instance for each new downstream task (Houlsby et al., 2019), substantially increasing memory consumption as the number of tasks grows. To mitigate this issue, Parameter-Efficient Transfer Learning (PETL; Ding et al. (2023)) methods have been developed, which update only a small subset of parameters. Among PETL approaches, Prompt Tuning (Lester et al., 2021; Li & Liang, 2021) that updates only a set of learnable vectors prepended to the layer inputs, has attracted considerable attention due to its strong performance and high parameter efficiency.

Although prompt tuning demonstrates remarkable efficiency-efficacy trade-off, it still faces two key challenges: (i) high sensitivity to prompt initialization (Wang et al., 2024b), and (ii) performance degradation under data scarcity (Su et al., 2022). To address these challenges, prompt transfer (Vu et al., 2022) is explored as a promising solution which pretrains source prompts on data-rich tasks and uses them to initialize target prompts. However, conventional prompt transfer assumes that the source and target tasks share the same modality, which limits its applicability when no modality-compatible source task is available. To overcome this limitation, Cross-Modality Prompt Transfer (CMPT; Zhang et al. (2025)) has been proposed to eliminate the modality constraint and improve flexibility. CMPT enables knowledge transfer across different modalities, allowing data-scarce tasks to benefit from the abundant data resources of data-rich modalities.

Previous work (Zhang et al., 2025) has verified the feasibility and provided a straightforward projection transfer approach for CMPT. However, its exploration was limited to **shallow** architectures, leaving open questions about the optimal strategy for transferring cross-modality knowledge to deep architectures. Investigating deep architectures in the cross-modality setting is essential, as they offer greater expressive capacity (Petrov et al., 2024a;b) and typically require shorter prompt lengths: an important structural advantage for enhancing transfer performance and accelerating inference

in cross-modality prompt transfer. On the other hand, the straightforward projection transfer approach, although simple and effective, transfers **all** source prompt vectors to the target task without addressing knowledge redundancy or compatibility issues. In cross-modality scenarios, substantial discrepancies between the source and target tasks can render certain vectors uninformative or even detrimental. Naïvely transferring all vectors may therefore hinder transfer performance.

This work focuses on developing a target-oriented, deep-architectural prompt transfer method. Our study is motivated by a key hypothesis: ***pre-trained source prompt exhibits redundancy and lack layer-wise compatibility in cross-modality prompt transfer***. In other words, not all prompt vectors are beneficial for a given target task. Moreover, the most transferable prompt vectors for a target layer may originate from arbitrary source layers, rather than strictly corresponding ones.

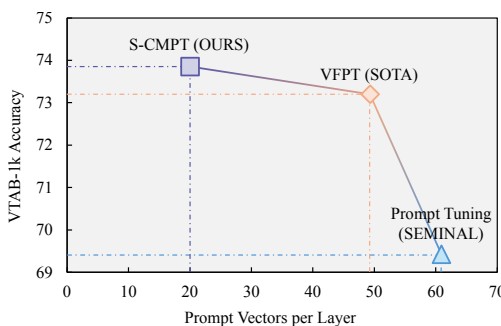

Figure 1: **X Axis: Prompt Vectors per Layer** (Averaged on VTAB-1k (Zhai et al., 2019)). **Y Axis: Average Performance on VTAB-1k**. Our proposed S-CMPT achieves the best performance with the fewest prompt vectors.

Based on our hypothesis and empirical observations, we propose **S**elective **C**ross-**M**odality **P**rompt **T**ransfer (**S-CMPT**), a method that first automatically selects transferable prompt vectors and then adapts them through a single-layer linear projection. Specifically, for each target layer, S-CMPT employs a learnable attention-based prompt selector to identify the most relevant source vectors. We further introduce a novel diversity regularizer that promotes the selection of non-redundant and complementary vectors. The selected vectors are subsequently adapted to the target task via a single-layer linear transformation. S-CMPT offers the following key advantages:

1. **Novelty.** S-CMPT is the first method specifically designed for deep prompt architectures in cross-modality prompt transfer. It challenges the conventional assumption that prompt vectors pretrained for the $i$-th layer must be transferred exclusively to the $i$-th layer of the target model. Instead, S-CMPT introduces a simple yet effective prompt selection strategy that enables flexible, layer-agnostic reuse of source prompt vectors, enhancing adaptability and transferability across modalities.

2. **Simplicity.** S-CMPT maintains simplicity throughout implementation and training. It only relies on fundamental modules (e.g., linear layers) and basic operations (e.g., matrix multiplication), avoiding architectural complexity. Moreover, it eliminates the need for exhaustive prompt-length tuning; a fixed length of 20 is sufficient to achieve strong performance.

3. **Efficiency & Efficacy.** As demonstrated in Figure 1, S-CMPT maintains excellent efficiency-efficacy trade-off: it achieves a +4.42% accuracy gain over prompt tuning and a +0.65% improvement over state-of-the-art prompt method (Zeng et al., 2024), while requiring only ∼40% of the prompt vectors compared to baseline approaches.

## 2 RELATED WORKS

### 2.1 PROMPT TUNING

Prompt tuning (Lester et al., 2021; Li & Liang, 2021; Wang et al., 2023) is a prominent class of Parameter-Efficient Transfer Learning (PETL) methods (Ding et al., 2023; Mai et al., 2024), which adapt pretrained transformers using learnable vectors prepended to the layer inputs. Originally developed for the language domain, Decomposed Prompt Tuning (Shi & Lipani, 2024) was proposed to address the challenge of long input sequences by decomposing the prompt into a shorter one and a pair of low-rank matrices that update word embeddings. In the vision domain, Jia et al. (2022) pioneered its application by introducing Visual Prompt Tuning (VPT) for Vision Transformers (ViT; Dosovitskiy et al. (2021)). Subsequent advancements include $E^2$VPT (Han et al., 2023), which injects prompts into both the query and key matrices, and the state-of-the-art Visual Fourier Prompt

Tuning (VFPT; Zeng et al. (2024)), which incorporates both spatial- and frequency-domain information into the prompt tuning process. Prompt tuning has also been extended to the vision–language domain to adapt vision-language models (VLMs). Multimodal Prompt Tuning (Wang et al., 2024a) integrates learnable prompts into both the vision encoder and the language model to enhance modality alignment and zero-shot instruction following. Similarly, to adapt VLMs in federated learning (FL) settings, FedMVP (Singha et al., 2025) addresses poor generalization in FL by generating dynamic visual prompts conditioned on multimodal contextual information. PromptKD (Li et al., 2024) presented an unsupervised framework that employs prompts to distill knowledge from a large teacher VLM to a lightweight student model. In our experiments, to ensure a fair comparison, we use prompt tuning methods specifically designed for the vision domain as our baselines, as our S-CMPT is closely related to them both architecturally and in terms of task formulation.

## 2.2 PROMPT TRANSFER

While parameter-efficient, prompt tuning typically underperforms full finetuning and suffers from slower convergence and optimization instability (Li & Liang, 2021). Prompt transfer addresses these limitations by pretraining prompts on data-rich tasks and using them as prompt initialization for downstream tasks. The seminal work conducted by Vu et al. (2022) first explored prompt transfer across different language tasks. Subsequently, Su et al. (2022) extended prompt transfer to cross-model scenarios. Zhang et al. (2025) pioneered Cross-Modality Prompt Transfer (CMPT), demonstrating that prompts pretrained on linguistic tasks can benefit data-scarce visual tasks via a learnable linear projection. Building on the existing foundation, we extend CMPT from shallow to deep architectures and propose S-CMPT that identifies transferable prompt vectors before projecting them into the target model.

## 3 METHODOLOGY

This section begins by introducing the preliminary concepts of Cross-Modality Prompt Transfer and presenting empirical findings from our pilot experiments that motivate our method. We then detail our proposed **Selective Cross-Modality Prompt Transfer** in Section 3.2.

### 3.1 CROSS-MODALITY PROMPT TRANSFER

Cross-Modality Prompt Transfer (CMPT) alleviates data scarcity in downstream tasks by transferring source prompts pretrained on data-rich language tasks. Prior work on CMPT has primarily focused on shallow architectures (Zhang et al., 2025). In this work, we explore CMPT in deep architectures, with the structure illustrated in Figure 3a. CMPT consists of two sequential phases:

1. *Pretraining*: The source prompt $p_s$ is pretrained on a data-rich NLP task using a pretrained language model $h_s$. The resulting source prompt has dimensionality $(N_l, N_p, d_h)$, consisting of $N_l$ layers with $N_p$ prompt vectors per layer, each of dimension $d_h$. Here, $N_l$ and $d_h$ denote the number of transformer layers and hidden dimension of $h_s$, respectively.

2. *Transfer*: The pretrained source prompt $p_s$ is adapted to the target task and model $h_t$ through $N_l$ learnable linear projectors $\{P^{(i)}\}_{i=1}^{N_l}$.

The training objective of CMPT is formalized as:

$$Z^{(1)} = h_t^{(1)} \left( \left[ P^{(1)}(p_s^{(1)}), X_t \right] \right)$$

$$Z^{(i)} = h_t^{(i)} \left( \left[ P^{(i)}(p_s^{(i)}), Z^{(i-1)} \right] \right)$$

$$\arg\min_{P} \mathcal{L} \left( \text{Logit}(Z^{(N_l)}), Y_t \right) \tag{1}$$

where $[\cdot, \cdot]$ denotes vector concatenation; $h_t^{(i)}$ and $p_s^{(i)}$ represent the $i$-th layer of $h_t$ and $p_s$ respectively; $P^{(i)}$ is the $i$-th linear projector; $Z^{(i)}$ is the hidden state output of $h_t^{(i)}$; $X_t$ and $Y_t$ are the target task inputs and labels; $\text{Logit}$ denotes the conversion of hidden states to probability logits.

CMPT operates under a layer-wise correspondence assumption, where each $p_s^{(i)}$ is transferred exclusively to the corresponding target layer $h_t^{(i)}$. However, this rigid alignment may be suboptimal due to the significant modality gap between source and target tasks. We hypothesize that for $h_t^{(i)}$, the performance can be improved by selecting $N_p$ vectors from the *entire* source prompt $p_s$, rather than limiting the selection to $p_s^{(i)}$. Based on this hypothesis, we conduct a pilot experiment involving the following two steps:

1. *Random Selection*: Treat $p_s$ as a global vector pool and randomly select $N_p$ vectors per target layer. Selection is done with replacement, allowing some vectors to be sampled multiple times while others may not be selected.
2. *Transfer*: Apply standard CMPT on VTAB-CIFAR using the randomly selected vectors.

We repeat this *random-select* procedure across 36 different random seeds and compare the test accuracy to that of CMPT. As shown in Figure 2, the raincloud plot illustrates a striking outcome: **Random selection outperforms CMPT in 34 out of 36 trials (94.4%), indicating that the pretrained source prompt exhibits redundancy and strict layer-wise correspondence hampers transfer performance** (the statistical significance will be provided in the supplementary material) . These findings motivate the need for a principled prompt selection mechanism that can more effectively identify transferable prompt vectors in cross-modality scenarios.

### 3.2 Selective Cross-Modality Prompt Transfer

**Overview.** Motivated by the findings, we propose **S**elective **C**ross-**M**odality **P**rompt **T**ransfer (**S-CMPT**), a simple yet effective approach that incorporates an attention-based selection mechanism prior to transfer. S-CMPT significantly enhances performance while preserving parameter efficiency. As illustrated in Figure 3b&c, S-CMPT operates as follows: at the $i$-th layer of the target model, a prompt selector $S^{(i)}$ identifies $N_p$ vectors from the source prompt $p_s$. The selected vectors $\hat{p}_s^{(i)}$ are then passed through a linear projector $P^{(i)}$ and prepended to the input embeddings of layer $i$ in the target model. To encourage informative and non-redundant selection, we incorporate a diversity regularization term $\mathcal{L}_{\text{div}}$ into the training objective, which discourages the selector from re-

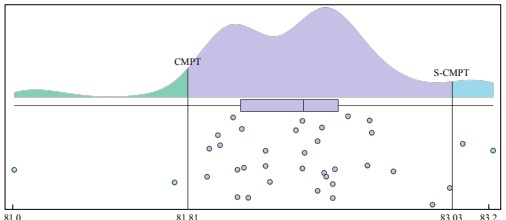

Figure 2: **Performance distribution of *random-select* runs on VTAB-CIFAR.** Dots represent individual runs (36 seeds). Vertical lines mark the best results obtained by CMPT (81.81%) and S-CMPT (83.03%).

peatedly choosing similar vectors. Crucially, during training, only the selectors $S$ and projectors $P$ are optimized. After training, these modules are discarded, and only the selected and projected prompts are retained for inference. This design substantially reduces storage requirements, adhering to the core goal of parameter-efficient transfer learning (Ding et al., 2023).

**Prompt Selector.** The prompt selector is designed to dynamically select $N_p$ vectors from the source prompt $p_s$ based on the needs of specific target tasks and transformer layers. Different downstream tasks and model layers exhibit diverse characteristics, necessitating tailored knowledge transfer. Furthermore, complex tasks may benefit from synthesizing information across multiple prompt vectors, even from different layers. To meet these challenges, the prompt selector must: (1) adapt to task and layer-specific requirements, and (2) aggregate information across the entire source prompt. The attention mechanism (Bahdanau et al., 2015) naturally satisfies these properties: it is data-adaptive and computes output vectors as weighted combinations of input vectors. We therefore implement the prompt selector using a simplified attention mechanism:

$$\hat{p}_s^{(i)} = S^{(i)}(p_s) = \text{softmax}\left(\frac{Q^{(i)}p_s^{\top}}{\tau}\right)p_s \qquad (2)$$

where $p_s \in \mathbb{R}^{(N_l \times N_p) \times d_h}$ denotes the flattened source prompt, and $Q^{(i)} \in \mathbb{R}^{N_p \times d_h}$ is a learnable query matrix for the $i$-th layer. Each row of $Q^{(i)}$ corresponds to a prompt vector to be selected. By

Figure 3: **(a) Vanilla CMPT** transfers layer-specific prompts via individual projectors. **(b) S-CMPT** follows a *select-then-transfer* paradigm. **(c) Prompt selector** employs attention operations for vector selection. Modules marked with 🔥 are trainable.

adjusting the number of queries in $Q^{(i)}$, we can flexibly determine the number of prompt vectors at each target layer. The temperature parameter $\tau$ controls the sharpness of the attention distribution.

**Diversity Regularization.** To mitigate redundancy in the source prompt and enhance task adaptability, we introduce a diversity regularization term:

$$\mathcal{L}_{\text{div}} = \sum_{i=1}^{N_l} \frac{1}{N_p(N_p - 1)} \sum_{m=1}^{N_p} \sum_{\substack{n=1 \\ n \neq m}}^{N_p} \cos\left(\hat{p}_{s,m}^{(i)}, \hat{p}_{s,n}^{(i)}\right) \quad (3)$$

where $cos$ denotes the cosine similarity operation and $\hat{p}_{s,n}^{(i)}$ denotes the n-th selected prompt vector at layer i. This regularization is motivated by two key considerations:

1. It encourages orthogonality among the selected vectors, thereby maximizing their representational diversity. This is particularly important because source prompts may contain semantically redundant vectors, which can degrade the information capacity.

2. It promotes the use of distinct vectors by enforcing more diverse attention distributions, which can help sample complementary information for the target task.

Following Petrov et al. (2024a), we provide a theoretical analysis in our supplementary material to show that enforcing $\mathcal{L}_{\text{div}}$ is equal to increasing the expressiveness of prompt tuning.

After incorporating diversity regularization, the overall training objective of S-CMPT becomes:

$$Z^{(1)} = h_t^{(1)}\left(\left[P^{(1)}\left(S^{(1)}(p_s)\right), X_t\right]\right)$$

$$Z^{(i)} = h_t^{(i)}\left(\left[P^{(i)}\left(S^{(i)}(p_s)\right), Z^{(i-1)}\right]\right)$$

$$\underset{P,S}{\arg\min}\,\mathcal{L}\left(\text{Logit}\left(Z^{(N_l)}\right), Y_t\right) + \lambda\mathcal{L}_{\text{div}} \quad (4)$$

where $\lambda$ controls regularization strength and is determined empirically. Note that $S^{(i)}$ selects from the full source prompt $p_s$ rather than a layer-specific subset.

## 4 EXPERIMENTS

### 4.1 EXPERIMENTAL SETUP

**Source Prompt Pretraining.** Following the findings of Zhang et al. (2025) that more source data yields better transfer performance, we pretrain the source prompt $p_s$ on SNLI (Bowman et al., 2015) using a RoBERTa-base model (Liu et al., 2019). To facilitate cross-modality transfer, RoBERTa is selected as the source model due to its architectural similarity to our target model: ViT (Dosovitskiy et al., 2021). Note that the source prompt is trained using a deep prompt tuning architecture. More technical details of pretraining the source prompt can be found in the supplementary material.

Table 1: **Image classification accuracy** on **VTAB-1k** using a **ViT-Base/16** backbone pretrained on **supervised ImageNet-21k**. [·] denotes the number of datasets within each group. *Acc.* denotes average accuracy (↑ means higher is better); $N_p$ indicates the average number of prompt vectors per layer (↓ means fewer is better). (·) refers to the number of wins compared to full finetuning (*Full*). Best results are in **bold**, second-best results are underlined. The results of full finetuning and VPT are obtained from the original paper of Visual Prompt Tuning. The results of VFPT and other baselines are obtained from the original paper of Visual Fourier Prompt Tuning. * means that the results of CMPT and S-CMPT are obtained through our experiments by averaging over three random seeds. The original CMPT only explored the shallow prompt architecture, it is re-implemented and rerun in the deep architecture in this work.

| | Natural [7] | | Specialized [4] | | Structured [8] | | *Overall* | |
|---|---|---|---|---|---|---|---|---|
| | Acc. (↑) | $N_p$ (↓) | Acc. (↑) | $N_p$ (↓) | Acc. (↑) | $N_p$ (↓) | Acc. (↑) | $N_p$ (↓) |
| *Full* | 75.88 | - | 83.36 | - | 47.64 | - | 65.57 | - |
| Bias | $73.30_{(3)}$ | - | $78.25_{(0)}$ | - | $44.09_{(2)}$ | - | 62.05 | - |
| Adapter | $70.67_{(4)}$ | - | $77.80_{(0)}$ | - | $33.09_{(0)}$ | - | 62.41 | - |
| LoRA | $78.26_{(5)}$ | - | $83.78_{(2)}$ | - | $56.20_{(7)}$ | - | 72.25 | - |
| VPT | $78.48_{(6)}$ | 12.4 | $82.43_{(2)}$ | 52.8 | $54.98_{(8)}$ | 107.5 | 69.43 | 60.9 |
| VFPT | $81.35_{(6)}$ | **9.9** | $84.93_{(4)}$ | 29.5 | $60.19_{(8)}$ | 93.8 | 73.20 | 49.3 |
| CMPT* | $81.48_{(6)}$ | 20 | $85.55_{(4)}$ | **20** | $60.44_{(8)}$ | **20** | 73.47 | **20** |
| S-CMPT* | $\mathbf{81.84}_{(6)}$ | 20 | $\mathbf{85.73}_{(4)}$ | **20** | $\mathbf{60.92}_{(8)}$ | **20** | **73.85** | **20** |

**Target Datasets.** We evaluate our method on the VTAB-1K (Zhai et al., 2019) benchmark, which includes 19 diverse image classification tasks categorized into three groups: (1) *Natural* tasks that contain natural images captured using standard cameras; (2) *Specialized* tasks that contain images captured using specialized equipment (e.g., medical or satellite imaging); (3) *Structured* tasks that require geometric understanding (e.g., distance estimation or object counting). Each dataset contains exactly 1000 training images, simulating data-scarce conditions and providing an ideal testbed for evaluating cross-modality prompt transfer methods. The acquisition, details, and pre-processing steps of the datasets will be elaborated in the supplementary material.

**Baselines.** We compare our proposed S-CMPT against the following baselines: (1) Full fine-tuning (*Full*) that retrains the entire model; (2) Classic parameter-efficient transfer learning approaches: Bias (Rebuffi et al., 2017), Adapter (Houlsby et al., 2019), and LoRA (Hu et al., 2022); (3) Vanilla Visual Prompt Tuning with deep architecture (VPT; Jia et al. (2022)); (4) The state-of-the-art prompt-based method, Visual Fourier Prompt Tuning (VFPT; Zeng et al. (2024)). All experiments are conducted using Vision Transformer base models with a patch size of 16 (ViT-B/16; Dosovitskiy et al. (2021)) and different pretraining objectives: (1) Supervised pretraining on ImageNet-21k (Deng et al., 2009); (2) Self-supervised pretraining on ImageNet-1k using MAE (He et al., 2022) and MoCo v3 (Chen et al., 2021).

**Hyperparameters.** Each VTAB-1k training set is split into 800 training and 200 validation samples for hyperparameter tuning. A grid search is performed over the following ranges: (1) Learning rates: {0.5, 0.1, 0.05, 0.01, 0.005, 0.001, 0.0005, 0.0002, 0.0001}; (2) Weight decay: {0, 0.0001, 0.001, 0.01, 0.1}. All models are trained for 100 epochs with a batch size of 64. The learning rate is warmed up over the first 10 epochs and then decayed to zero using a cosine schedule.

**Reproducibility.** To ensure reproducibility, we implement S-CMPT using the publicly available deep learning framework PyTorch (Paszke et al., 2019). We follow PyTorch's official guidelines to fix random seeds for each run and minimize nondeterminism. All experiments are conducted on NVIDIA RTX A5500 GPUs with 24GB of memory. Our implementation and codebase will be made publicly available to facilitate further research and verification.

Table 2: **Image classification accuracy** on **VTAB-1k** with a **ViT-Base/16** backbone pretrained on **ImageNet-1k** using self-supervised objectives: **MAE** and **MoCo v3**.

| | MAE | | | | MoCo v3 | | | |
|---|---|---|---|---|---|---|---|---|
| | Natural | Specialized | Structured | *Overall* | Natural | Specialized | Structured | *Overall* |
| *Full* | **59.31** | 79.68 | 53.82 | **61.29** | 71.95 | 84.72 | 51.98 | 66.22 |
| Bias | $54.55_{(1)}$ | $75.68_{(1)}$ | $47.70_{(0)}$ | 56.11 | $72.89_{(3)}$ | $81.14_{(0)}$ | $53.43_{(4)}$ | 66.43 |
| Adapter | $54.90_{(3)}$ | $75.19_{(1)}$ | $38.98_{(0)}$ | 52.47 | $74.19_{(4)}$ | $82.66_{(1)}$ | $47.69_{(2)}$ | 64.82 |
| VPT | $36.02_{(0)}$ | $60.61_{(1)}$ | $26.57_{(0)}$ | 37.22 | $70.27_{(4)}$ | $83.04_{(0)}$ | $42.38_{(0)}$ | 61.22 |
| VFPT | $53.59_{(6)}$ | $77.75_{(1)}$ | $36.15_{(1)}$ | 51.33 | $77.47_{(5)}$ | $85.76_{(3)}$ | $58.74_{(6)}$ | 71.33 |
| CMPT* | $55.39_{(4)}$ | $81.37_{(2)}$ | $49.77_{(3)}$ | 58.49 | **$78.18_{(6)}$** | $86.14_{(4)}$ | $60.69_{(7)}$ | 72.49 |
| S-CMPT* | $56.06_{(4)}$ | **$81.87_{(4)}$** | **$54.72_{(4)}$** | 60.93 | $78.04_{(6)}$ | **$86.36_{(4)}$** | **$61.12_{(7)}$** | **72.67** |

## 4.2 MAIN RESULTS

Table 1 presents the classification results using ViT-Base/16 pretrained on ImageNet-21k (supervised). Table 2 shows performance with self-supervised ViTs pretrained on ImageNet-1k via MAE and MoCo v3. From the information provided in the tables, we draw the following observations:

**Overall Performance.** S-CMPT consistently outperforms all baselines across all task types, including conventional classification and geometrically complex tasks. With supervised ViT, it achieves a gain of +4.42% over VPT, +0.65% over VFPT, and +0.38% over CMPT. This advantage is further magnified with self-supervised ViTs: +9.60% and +1.34% improvements over VFPT for MAE and MoCo, respectively; +2.44% and +0.18% over CMPT under the same settings. When setting *Full* as the comparison baseline, S-CMPT outperforms *Full* on 18 out of 19 datasets with ViT-supervised, matching the best prompt-based baseline VFPT. When the ViT is pretrained with MAE, S-CMPT surpasses *Full* on 12 datasets, compared to 8 datasets for VFPT. Under MoCo v3 pretraining, S-CMPT outperforms *Full* on 17 datasets, whereas VFPT does so on 14 datasets. These results validate two key conclusions:

1. **Cross-modality knowledge transfer is both important and effective**: On data-scarce tasks, by transferring source prompts pretrained on large-scale language datasets, both CMPT and S-CMPT surpass non-transfer prompt tuning methods. The source prompt serves as a rich knowledge bank, compensating for target data scarcity by providing transferable knowledge representations.

2. **Prompt selection is crucial for effective transfer**: S-CMPT outperforms CMPT by **+0.38%**, **+2.44%**, and **+0.18%** in overall accuracy across the supervised, MAE, and MoCo objectives respectively. These gains validate our hypothesis that source prompts exhibit redundancy and that strict layer-wise correspondence is suboptimal. The attention-based selection mechanism in S-CMPT can effectively prune irrelevant vectors and realign prompt knowledge to suit the specific demands of the target transformer and task.

**Memory Efficiency.** Despite achieving state-of-the-art performance, S-CMPT offers substantially higher parameter efficiency. By selecting only 20 prompt vectors per layer (a total of 240 vectors per task), S-CMPT uses approximately **2.5× fewer prompt vectors** than VFPT, which averages 49.3 prompt vectors per layer on the supervised ViT. This fixed-length design eliminates the need for exhaustive prompt length tuning, reducing both computational overhead and memory footprint, thereby making S-CMPT particularly suitable for resource-constrained scenarios.

**Performance on *Distant* Tasks.** Following the definitions in Han et al. (2024) and Zeng et al. (2024), we consider *distant tasks* within VTAB-1k to be those that differ significantly from the pretraining task (ImageNet-21k) in either data distribution or task objective. For example, SVHN is deemed distant due to its substantial distributional shift from ImageNet-21k, as quantified by the Fréchet Inception Distance (FID; Chong & Forsyth (2020); Kynkäänniemi et al. (2023)). Similarly, Kitti is categorized as distant because its objective (object distance estimation) significantly differs from the standard classification task. Empirically, we observe that S-CMPT offers significantly greater improvements on distant tasks. When using the supervised ViT, S-CMPT surpasses VPT by

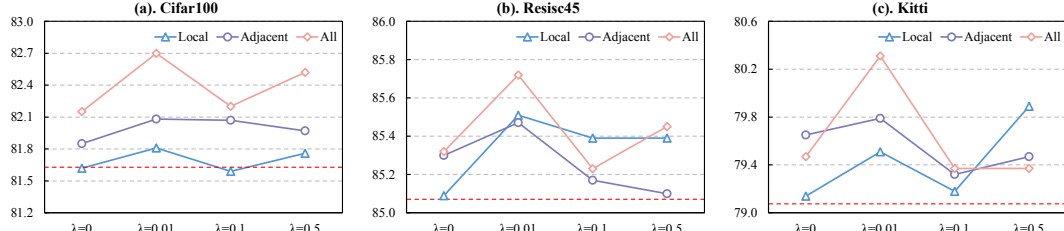

Figure 4: **Ablation Study** Conducted on CIFAR, Resisc45, and Kitti. The x axis stands for different $\lambda$ values (different strengths on diversity regularization). The y axis is the accuracy. The red dotted line represents the performance of CMPT (without selection and diversity regularization).

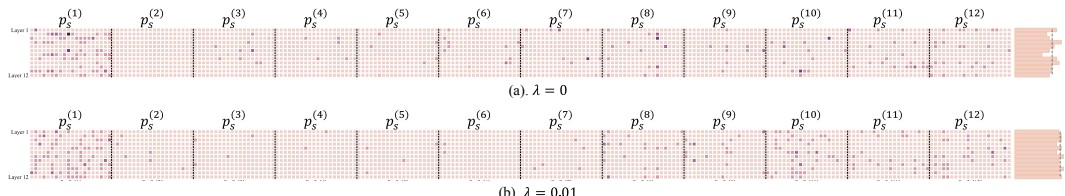

Figure 5: **Aggregated Attention Distribution of the Prompt Selectors**. Columns: source prompt vectors. Rows: query vectors in prompt selectors across layers. Matrix intensity: cumulative attention scores (Eq. 5). Right histograms: count of vectors with attention $> 0.5$ per layer, serving as a quantitative evidence of the diversity regularization being able to help the prompt selector attend to more vectors instead of repeatedly selecting from a small range of vectors.

+5.96% on distant tasks (SVHN, Patch Camelyon, Diabetic Retinopathy, and all VTAB-1k structured tasks). This gain notably exceeds the +2.31% improvement observed on non-distant tasks. We hypothesize that this discrepancy arises because distant tasks present substantial gaps in either visual semantics or task requirements compared to the pretraining task (ImageNet-21k). While conventional VPT struggles to bridge this gap, S-CMPT compensates by injecting filtered, task-relevant linguistic knowledge learned from source prompt pretraining. This additional cross-modality information enriches the ViT's representations, enabling better adaptation to tasks far removed from its original training distribution or objective.

The per-task accuracy, hyperparameters, and error bars are listed in the supplementary material.

### 4.3 ABLATION STUDIES

In this section, we conduct ablation studies to investigate: (1) the impact of the prompt vector selection range, and (2) the effect of the diversity regularization strength $\lambda$.

For (1), we design three vector selection range configurations:

1. **LOCAL**: The i-th prompt selector $S^{(i)}$ selects only from $p_s^{(i)}$ (i.e., the $i$-th layer of the source prompt).

2. **ADJACENT**: $S^{(i)}$ selects from $\{p_s^{(i-1)}, p_s^{(i)}, p_s^{(i+1)}\}$. Note that $p_s^{(i-1)}$ or $p_s^{(i+1)}$ will not be included in edge cases (i.e., the first and last layers).

3. **ALL**: $S^{(i)}$ selects from the entire $p_s$ (default setting).

For (2), we vary $\lambda \in \{0, 0.01, 0.1, 0.5\}$ to control the strength of the diversity regularization. The results, shown in Figure 4, lead to the following conclusions:

1. **Universal Benefit of Prompt Selection**: Without diversity regularization ($\lambda = 0$), the LOCAL setting performs on par with CMPT, while ADJACENT and ALL outperform LOCAL

by a notable margin. When $\lambda > 0$, all configurations outperform CMPT. This consistent performance gain confirms the necessity of prompt selection for effective cross-modality transfer.

2. **Broader Selection Range Enhances Transfer Performance**: ADJACENT outperforms LOCAL in 8 out of 12 cases, while ALL surpasses both ADJACENT and LOCAL in 9 out of 12 cases, demonstrating that a broader selection range leads to better transferability. This aligns with our hypothesis: the optimal prompt vectors for $h_t^{(i)}$ may originate from *any* source layer, not just $p_s^{(i)}$. Therefore, we adopt the ALL configuration as the default for prompt selection.

3. **Effective and Robust Diversity Regularization**: In all cases, configurations with $\lambda > 0$ outperform those with $\lambda = 0$, demonstrating the effectiveness of diversity regularization in promoting the selection of more expressive and diverse prompt vectors. Notably, almost all curves (8 out of 9) peak at $\lambda = 0.01$. However, these peaks are typically close in performance to the second-best $\lambda$ values, indicating that the method is not overly sensitive to the specific value of $\lambda$. This showcases the robustness of S-CMPT: although $\lambda$ is a manually set hyperparameter, S-CMPT does not rely on the extensive search of it to guarantee performance. Instead, a fixed value (e.g., $\lambda = 0.01$) works well across all scenarios.

## 4.4 PROMPT SELECTION VISUALIZATION

To examine how diversity regularization influences the selection of source prompt vectors, we visualize the aggregated attention distributions between all query vectors $Q^{(i)}$ in $S^{(i)}$ and the source prompt $p_s$. Each row of the resulting attention matrix is computed as:

$$Row^{(i)} = \sum_{j=0}^{N_p} \text{softmax}\left(\frac{Q_j^{(i)} p_s^\top}{\tau}\right) \tag{5}$$

where $Q_j^{(i)}$ denotes the j-th query vector in the prompt selector $S^{(i)}$, and $p_s$ is the flattened source prompt. Figure 5 presents these attention matrices, alongside histograms that quantify the number of $p_s$ vectors receiving attention scores greater than 0.5 per layer (evaluated on CIFAR-100).

From the figure, we draw the following observations:

1. **Diversity Regularization Enables Varied Selection Patterns**: The attention patterns differ significantly between the $\lambda = 0$ and $\lambda = 0.01$ configurations, suggesting that the diversity regularization term promotes more varied and flexible selection strategies, breaking away from fixed or repetitive attention modes.

2. **Redundancy Reduction**: In the absence of regularization ($\lambda = 0$), the attention matrix displays highly concentrated dark cells, indicating repeated selection of the same vectors (i.e., high redundancy). By contrast, with diversity regularization, the selector attends to a broader range of vectors, as reflected in the histogram, which shows a more distributed selection pattern.

These findings offer a deeper understanding of the prompt selection mechanism in S-CMPT, emphasizing the effectiveness of the diversity term in producing more diverse and less redundant prompt representations.

## 5 CONCLUSIONS

This work addresses two critical limitations in the field of cross-modality prompt transfer: (1) the restriction to shallow architectures, and (2) the naive transfer of redundant source prompt vectors. Through rigorous empirical analysis and method design, we establish three key contributions:

1. **Validation of the Layer Correspondence and Source Prompt Redundancy Hypothesis**: We empirically demonstrate: (1) Rigid layer-wise correspondence between source and target models is suboptimal for cross-modality prompt transfer. (2) The pretrained source

prompt exhibit redundancy and not all vectors are necessary for a target task. Our extensive experiments show that target-oriented prompt selection significantly outperforms fixed correspondence, challenging conventional assumptions.

2. **Selective Transfer Framework**: We introduce *Selective Cross-Modality Prompt Transfer* (S-CMPT), the first solution tailored for deep architectures in cross-modality prompt transfer. S-CMPT employs an attention-based selection mechanism to identify the most relevant vectors from all vectors in the source prompt. In addition, a diversity regularization term is incorporated to reduce redundancy and enhance expressiveness.

3. **Superior Parameter-Performance Trade-off**: On the VTAB-1k benchmark, S-CMPT achieves state-of-the-art performance (73.85% overall accuracy) using only 20 prompt vectors per layer, which is $2.5\times$ fewer than prior approaches. This highlights its optimal balance between representational power and parameter efficiency.

Overall, S-CMPT establishes a new paradigm for efficient knowledge transfer across modalities and architectures, demonstrating that *selective* knowledge adoption is more effective than indiscriminate prompt reuse.

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

# Supplementary Material

## A  LLM Usage

The authors acknowledge the use of ChatGPT for proofreading the paper (including grammar and spelling check) only.

## B  Datasets

To acquire VTAB-1k, we use the script provided by SPT (Sensitivity-Aware Visual Parameter-Efficient Tuning; ICCV2023) to download all the datasets in the format of PNG images.

The information of each dataset is listed in Table 3.

Table 3: **Information of VTAB-1k datasets.** When grid searching, the model is trained on the 800 training samples and evaluated on the 200 validation samples. When training, the model is trained on the 800+200 (training and validation) samples and evaluated on the test samples.

| Group | Dataset | Classes | Train | Val | Test |
|---|---|---|---|---|---|
| Natural | CIFAR-100 | 100 | | | 10,000 |
| | Caltech101 | 102 | | | 6,084 |
| | DTD | 47 | | | 1,880 |
| | Flowers102 | 102 | 800 | 200 | 6,149 |
| | Pets | 37 | | | 3,669 |
| | SVHN | 10 | | | 26,032 |
| | SUN397 | 397 | | | 21,750 |
| Specialized | Patch Camelyon | 2 | | | 32,768 |
| | EuroSAT | 10 | 800 | 200 | 5,400 |
| | Resisc45 | 45 | | | 6,300 |
| | Diabetic Retionpathy | 5 | | | 42,670 |
| Structured | DmLab | 6 | | | 22,735 |
| | Kitti | 4 | | | 711 |
| | SmallNORB_azi | 18 | | | 12,150 |
| | SmallNORB_ele | 9 | 800 | 200 | 12,150 |
| | dSprites_loc | 16 | | | 73,728 |
| | dSprites_ori | 16 | | | 73,728 |
| | Clevr_dist | 6 | | | 15,000 |
| | Clevr_count | 8 | | | 15,000 |

Regarding the image augmentation strategies for VTAB-1k, we follow the default settings in VTAB1K and do not use any augmentation tricks except the following:

1. Resizing the images to a size of $224 \times 224$;

2. Converting the images to PyTorch tensors and re-scaling them to $0 \sim 1$;

3. Normalizing the images using the mean and standard deviation values calculated from ImageNet ($mean = (0.485, 0.456, 0.406)$, $std = (0.229, 0.224, 0.225)$).

## C  Technical Details

### C.1  Source Prompt Pretraining

The deep-architectural source prompt is pretrained on the SNLI natural language inference dataset with a RoBERTa-base pretrained language model. RoBERTa-base has nearly the same architecture as ViT (i.e., 12 layers with a hidden dimension of 768), eliminating any dimensional barriers arising from the architectural differences of the source and target models.

**Input Embedding Structures.** Each sample of SNLI contains two sentences. When conducting deep prompt tuning on SNLI, the input embedding structure in Table 4 is adopted.

Table 4: The input embedding structure for training deep prompts on SNLI. The first row shows the order of different tokens. S1 and S2 refer to the first and second sentence, respectively. The second row indicates the number of each token. The numbers of S1 and S2 are omitted as they vary from sample to sample. The third row indicates whether the tokens are trainable. $T$ indicates trainable. The fourth row indicates whether the positional embeddings are added to the corresponding token. $P$ means added.

| [Prompt] | [MASK] | [CLS] | [S1] | [SEP] | [S2] | [SEP] |
|----------|--------|-------|------|-------|------|-------|
| 20 | 1 | 1 | | 1 | | 1 |
| T | × | × | × | × | × | × |
| × | P | P | P | P | P | P |

**Output Feature Pooling.** To deal with the features produced by the RoBERTa backbone, we simply take the feature vector corresponding to the [MASK] token and pass it to RoBERTa's pretrained language modeling head to obtain the logits. What follows is loss calculation given the ground-truth labels.

**Hyperparameters.** The hyperparameters for pretraining the source prompt on SNLI with RoBERTa is outlined in Table 5:

Table 5: The hyperparameters for pretraining the source prompt

| Epoches | LR | LR Decay? | LR Warm-Up | WD | Batch Size |
|---------|-----|-----------|------------|-----|------------|
| 64 | 0.001 | No | 0 | 0 | 16 |

## C.2 PROMPT TRANSFER TO TARGET TASKS

**Input Embedding Structures.** In our experiments, a uniform prompt length of 20 is employed and the input embedding structure demonstrated in Table 6 is adopted.

Table 6: The input embedding structure for ViT

| [CLS] | [Prompt] | [Image] |
|-------|----------|---------|
| 1 | 20 | 196 |
| × | × | × |
| P | × | P |

**Output Feature Pooling.** For ViT-supervised and ViT-MoCo, we use the [CLS] pooling method: pass the output feature vector corresponding to the [CLS] token to the classification head. For ViT-MAE, we take the average of the all the feature vectors (except the one corresponding to the [CLS] token) and pass the averaged vector to the classification head.

**Hyperparameters** The per-task hyperparameters of S-CMPT is listed in Table 7 (ViT-B/16 Supervised), Table 8 (ViT-B/16 MAE), and Table 9 (ViT-B/16 MoCo v3). Note that we use a universal batch size of 64 across all scenarios. The number of prompt vectors per layer is universally set as 20 across all scenarios. We do not use any prompt dropouts and set the value of drop path rate as 0 across all scenarios. The value of $\lambda$ that controls the strength of the diversity term is always 0.01. The only differences in the hyperparameters are the learning rate, weight decay, and the type of optimizers, as listed in the tables.

Table 7: Hyperparameters of S-CMPT with a supervised ViT.

| Dataset | LR | WD | Optimizer |
|---|---|---|---|
| CIFAR-100 | 0.005 | 0.001 | |
| Caltech101 | 0.005 | 0.001 | |
| DTD | 0.01 | 0.001 | |
| Flowers102 | 0.01 | 0.001 | Adam |
| Pets | 0.01 | 0.001 | |
| SVHN | 0.0005 | 0.1 | |
| SUN397 | 0.01 | 0.001 | |
| Patch Camelyon | 0.001 | 0.01 | |
| EuroSAT | 0.01 | 0.001 | Adam |
| Resisc45 | 0.005 | 0.001 | |
| Diabetic Retionpathy | 0.0005 | 0 | |
| DmLab | 0.001 | 0.01 | |
| Kitti | 0.001 | 0.1 | |
| SmallNORB_azi | 0.0005 | 0.001 | |
| SmallNORB_ele | 0.0002 | 0.1 | Adam |
| dSprites_loc | 0.0005 | 0.0001 | |
| dSprites_ori | 0.001 | 0.0001 | |
| Clevr_dist | 0.001 | 0.1 | |
| Clevr_count | 0.0001 | 0.001 | |

Table 8: Hyperparameters of S-CMPT with ViT-MAE

| Dataset | LR | WD | Optimizer |
|---|---|---|---|
| CIFAR-100 | 0.05 | 0.1 | |
| Caltech101 | 0.05 | 0.1 | |
| DTD | 0.001 | 0.0001 | |
| Flowers102 | 0.0005 | 0.1 | AdamW |
| Pets | 0.001 | 0 | |
| SVHN | 0.005 | 0.01 | |
| SUN397 | 0.05 | 0.1 | |
| Patch Camelyon | 0.0005 | 0 | |
| EuroSAT | 0.0005 | 0 | AdamW |
| Resisc45 | 0.0005 | 0.001 | |
| Diabetic Retionpathy | 0.005 | 0.001 | |
| DmLab | 0.01 | 0.01 | |
| Kitti | 0.0005 | 0.1 | |
| SmallNORB_azi | 0.0001 | 0.01 | |
| SmallNORB_ele | 0.0001 | 0.001 | AdamW |
| dSprites_loc | 0.0001 | 0 | |
| dSprites_ori | 0.0001 | 0.1 | |
| Clevr_dist | 0.0005 | 0.01 | |
| Clevr_count | 0.0005 | 0.1 | |

## D  PER-TASK PERFORMANCE

The per-task accuracy of full finetuning (*Full*), VFPT, CMPT, and S-CMPT is shown in Table 10. Note:

- The performance of VPT is omitted since VPT falls below a large margin compared to other baselines.

- The performance of *Full* is obtained from the original paper of VPT and VFPT.

- The performance of VFPT is obtained from its original paper.

Table 9: Hyperparameters of S-CMPT with ViT-MoCo

| Dataset | LR | WD | Optimizer |
|---|---|---|---|
| CIFAR-100 | 0.01 | 0.0001 | |
| Caltech101 | 0.01 | 0.0001 | |
| DTD | 0.05 | 0.0001 | |
| Flowers102 | 0.05 | 0 | Adam |
| Pets | 0.01 | 0.0001 | |
| SVHN | 0.05 | 0 | |
| SUN397 | 0.01 | 0 | |
| Patch Camelyon | 0.001 | 0.001 | |
| EuroSAT | 0.05 | 0.0001 | Adam |
| Resisc45 | 0.05 | 0.0001 | |
| Diabetic Retionpathy | 0.005 | 0 | |
| DmLab | 0.01 | 0.0001 | |
| Kitti | 0.0005 | 0.1 | |
| SmallNORB_azi | 0.005 | 0 | |
| SmallNORB_ele | 0.0005 | 0 | |
| dSprites_loc | 0.001 | 0.001 | Adam |
| dSprites_ori | 0.05 | 0 | |
| Clevr_dist | 0.005 | 0 | |
| Clevr_count | 0.001 | 0 | |

Table 10: **Detailed results on VTAB-1k.** Within each pretraining objective, the best results are **bold**. The first row of CMPT and S-CMPT is the mean accuracy while the second row is the standard deviation.

| | Method | Caltech101 | CIFAR100 | DTD | Flowers102 | Pets | Sun397 | SVHN | Overall. | Patch Camelyon | Resisc45 | EuroSAT | Retinopathy | Overall. | DMLab | KITTI | SmallNORB/azi | SmallNORB/ele | dSprites/loc | dSprites/ori | Clevr/dist | Clevr/count | Overall. |
|---|---|---|---|---|---|---|---|---|---|---|---|---|---|---|---|---|---|---|---|---|---|---|---|
| | | | | | *Natural* | | | | | | | *Specialized* | | | | | | *Structured* | | | | | |
| **SUP** | *Full* | 87.7 | 68.9 | 64.3 | 97.2 | 86.9 | 38.8 | 87.4 | 75.88 | 79.7 | 84.2 | 95.7 | 73.9 | 83.36 | 41.7 | 65.5 | 25.7 | 29.1 | 57.5 | 46.7 | 58.6 | 56.3 | 47.64 |
| | VFPT | 91.4 | 80.7 | 69.4 | 99.3 | 90.3 | 52.7 | 85.6 | 81.35 | 83.5 | 84.4 | 96.5 | 75.4 | 84.93 | 48.3 | 79.3 | **34.1** | **43.4** | 81.5 | **56.0** | 63.2 | 75.8 | 60.19 |
| | CMPT | 91.19 | 81.63 | 70.30 | 98.59 | 90.10 | 52.90 | 85.62 | 81.48 | 84.49 | 85.08 | **96.71** | 75.90 | 85.55 | 47.31 | 79.09 | 31.05 | 43.36 | **89.13** | 53.84 | 62.18 | **77.54** | 60.44 |
| | | 0.67 | 0.18 | 0.27 | 0.08 | 0.13 | 0.84 | 0.80 | | 0.30 | 0.30 | 0.16 | 0.20 | | 0.15 | 0.21 | 0.20 | 0.78 | 1.40 | 1.47 | 0.44 | 0.49 | |
| | S-CMPT | 91.37 | **82.70** | **70.80** | 98.59 | 90.10 | **53.42** | 85.87 | **81.84** | **84.61** | **85.72** | 96.59 | **76.00** | **85.73** | **48.48** | **80.31** | 31.77 | 43.19 | 88.87 | 55.21 | 62.20 | 77.36 | **60.92** |
| | | 0.21 | 0.25 | 0.55 | 0.03 | 0.35 | 0.25 | 0.16 | | 0.59 | 0.24 | 0.10 | 0.10 | | 0.43 | 0.86 | 0.88 | 1.24 | 0.26 | 0.59 | 0.21 | 0.54 | |
| **MAE** | *Full* | 84.2 | 24.6 | 56.9 | 72.7 | 74.4 | 15.8 | 86.6 | 59.31 | 81.8 | 72.3 | 94.0 | 70.6 | 79.68 | 45.2 | 75.3 | 30.2 | 33.0 | 72.5 | 47.5 | 59.8 | 67.0 | 53.82 |
| | VFPT | 87.7 | 36.0 | 58.0 | 74.3 | 76.3 | 23.3 | 19.6 | 53.59 | 76.9 | 69.2 | 91.3 | 73.6 | 77.75 | 40.7 | 80.7 | 9.3 | 17.3 | 13.7 | 34.6 | 45.3 | 47.6 | 36.15 |
| | CMPT | 80.91 | 32.18 | 59.02 | 73.27 | 49.34 | 16.56 | 76.47 | 55.39 | **86.07** | 71.1 | 93.86 | **74.45** | 81.37 | 42.9 | 81.95 | 5.83 | 23.09 | **85.43** | 35.09 | 61.58 | 62.26 | 49.77 |
| | | 6.94 | 0.94 | 1.53 | 0.7 | 25.54 | 1.54 | 1.05 | | 0.91 | 2.1 | 1.23 | 0.26 | | 2.83 | 1.06 | 0.15 | 6.18 | 1.69 | 22.66 | 1.78 | 13.36 | |
| | S-CMPT | 82.24 | 28.10 | **60.67** | **74.55** | 56.68 | 17.45 | 72.75 | 56.06 | 85.22 | **72.94** | **95.33** | 73.98 | **81.87** | 42.02 | **83.83** | 16.58 | 28.51 | 84.46 | 45.05 | **62.99** | **74.34** | **54.72** |
| | | 3.97 | 1.80 | 1.96 | 2.07 | 22.38 | 2.21 | 2.90 | | 1.33 | 1.44 | 0.59 | 0.36 | | 0.39 | 1.59 | 9.09 | 2.25 | 1.64 | 4.87 | 0.30 | 4.07 | |
| **MoCo** | *Full* | 91.0 | 57.6 | 64.6 | 91.6 | 79.9 | 29.1 | **89.8** | 71.94 | 85.1 | 83.1 | 96.4 | 74.2 | 84.70 | 44.6 | 77.9 | **31.5** | 36.9 | 63.8 | 49 | 56.9 | 55.2 | 51.98 |
| | VFPT | 90.5 | 73.6 | 70.5 | 92.4 | 88.3 | **42.3** | 84.7 | 77.47 | 86.7 | 85.2 | 95.7 | 75.5 | 85.76 | 46.1 | 82.2 | 23.8 | **45.8** | 85.3 | 47.4 | 63 | 76.3 | 58.74 |
| | CMPT | **91.79** | 76.35 | 70.8 | 92.57 | **89.45** | 40.57 | 85.74 | **78.18** | 86.33 | 85.47 | 96.94 | **75.8** | 86.14 | 49 | 82.47 | 28.44 | 44.27 | 85.39 | 52.28 | **63.82** | 79.88 | 60.69 |
| | | 0.21 | 0.14 | 0.11 | 0.08 | 0.26 | 1.41 | 0.53 | | 0.26 | 0.03 | 0.29 | 0.31 | | 0.27 | 0.21 | 0.77 | 2.61 | 1.6 | 1.23 | 0.32 | 1.52 | |
| | S-CMPT | 91.74 | **76.39** | **71.06** | **92.95** | 89.15 | 40.56 | 84.43 | 78.04 | **86.87** | **85.76** | **97.05** | 75.74 | **86.36** | **49.16** | **82.89** | 28.44 | 44.77 | **85.49** | **54.51** | 63.12 | **80.59** | **61.12** |
| | | 0.10 | 0.40 | 0.09 | 0.16 | 0.26 | 0.07 | 1.27 | | 0.36 | 0.33 | 0.14 | 0.19 | | 0.66 | 0.80 | 3.57 | 3.78 | 0.31 | 0.86 | 0.48 | 0.59 | |

- The performance of CMPT and S-CMPT is obtained from our experiments, averaging over three different random seeds: [42, 44, 100].

## E THEORETICAL ANALYSIS

In this section, we provide a theoretical justification for our diversity regularization, $\mathcal{L}_{\text{div}}$, demonstrating its role in enhancing the expressiveness of prompt tuning. Our analysis is based on the theoretical framework established by Petrov et al. (2024a), which formalizes the function of prompts as additive bias terms within an attention block. We adapt this framework to demonstrate how $\mathcal{L}_{\text{div}}$ directly addresses a key limitation in this bias's expressive capacity. For simplicity, we analyze a single-head attention module with query, key, and value projection matrices $W_q$, $W_k$, and $W_v$.

### E.1 ATTENTION WITHOUT PROMPT VECTORS

Let the input embeddings be represented by the matrix $X = [x_1, x_2, \ldots, x_n] \in \mathbb{R}^{d_h \times n}$, where $d_h$ is the hidden dimension and $n$ is the sequence length. The resulting attention matrix $A \in \mathbb{R}^{n \times n}$ is:

$$A = \begin{bmatrix} A(x_1, x_1) & A(x_1, x_2) & \dots & A(x_1, x_n) \\ A(x_2, x_1) & A(x_2, x_2) & \dots & A(x_2, x_n) \\ \vdots & \vdots & \ddots & \vdots \\ A(x_n, x_1) & A(x_n, x_2) & \dots & A(x_n, x_n) \end{bmatrix} \qquad (6)$$

$A(x_i, x_j)$ is the scalar attention score from query $x_i$ to key $x_j$:

$$A(x_i, x_j) = \frac{e^{(W_q x_i)^T (W_k x_j)}}{\sum_{r=1}^{n} e^{(W_q x_i)^T (W_k x_r)}} \qquad (7)$$

where $W_q$ and $W_k$ are the pretrained query and key projection matrices of the attention module.

The output $o_i \in \mathbb{R}^{d_h}$ for the $i$-th token is a weighted sum of the value vectors:

$$o_i = A(x_i, x_1) W_v x_1 + A(x_i, x_2) W_v x_2 + \dots + A(x_i, x_n) W_v x_n \qquad (8)$$

where $W_v$ is the pretrained value projection matrix of the attention module.

## E.2 ATTENTION WITH PROMPT VECTORS

After prepending $N_p$ prompt vectors, the input embeddings become:

$$X' = [p_1, \dots, p_{N_p}, x_1, \dots, x_n] \in \mathbb{R}^{d_h \times (N_p + n)}$$

We analyze the attention outputs $o_i^p$ corresponding to the **original** $n$ input tokens (as in deep prompt architecture, the outputs corresponding to the prompt vectors are discarded at each layer). The attention matrix for these tokens are now $n \times (N_p + n)$:

$$\begin{bmatrix} A(x_1, p_1) & \dots & A(x_1, p_{N_p}) & A(x_1, x_1) & \dots & A(x_1, x_n) \\ A(x_2, p_1) & \dots & A(x_2, p_{N_p}) & A(x_2, x_1) & \dots & A(x_2, x_n) \\ \vdots & \ddots & \vdots & \vdots & \ddots & \vdots \\ A(x_n, p_1) & \dots & A(x_n, p_{N_p}) & A(x_n, x_1) & \dots & A(x_n, x_n) \end{bmatrix} \qquad (9)$$

The attention scores are scaled down by the presence of the prompt vectors:

$$A(x_i, x_j) = \frac{e^{(W_q x_i)^T (W_k x_j)}}{\sum_{r=1}^{N_p} e^{(W_q x_i)^T (W_k p_r)} + \sum_{r=1}^{n} e^{(W_q x_i)^T (W_k x_r)}} \qquad (10)$$

Consequently, the output $o_i^p$ for the $i$-th token is modified as follows:

$$o_i^p = \underbrace{A(x_i, p_1) W_v p_1 + \dots + A(x_i, p_{N_p}) W_v p_{N_p}}_{\text{Additive Prompt Bias } (\beta_i)} + A(x_i, x_1) W_v x_1 + \dots + A(x_i, x_n) W_v x_n \qquad (11)$$

Comparing Eq. 11 and Eq. 8, two primary modifications are evident:

1. An **additive bias term** $\beta_i$ (highlighted in red) is introduced to the output.
2. The original attention scores ($A(x_i, x_j)$, etc.) are re-normalized (scaled down) due to the new prompt-related terms in the softmax denominator (Eq. 10).

As noted in the seminal theoretical work (Petrov et al., 2024a), the prompt's primary influence is this additive bias term, $\beta_i$. This bias is a weighted sum of the projected prompt vectors, $\{W_v p_j\}_{j=1}^{N_p}$.

The expressive capacity of this bias is therefore bounded by the **span** of this set of projected vectors. Consider an extreme case where the selected prompt vectors $\{p_j\}$ are collinear (i.e., they form a rank-1 set). Assuming $W_v$ is not a degenerate mapping, the projected vectors $\{W_v p_j\}$ will also be collinear. Consequently, the bias term $\beta_i$ will be restricted to a single direction in the output space, regardless of the attention weights.

A set of prompt vectors $\{p_j\}$ with a larger span (i.e., higher effective rank) will result in a set of value vectors $\{W_v p_j\}$ that also has a larger span. This allows the additive bias term $\beta_i$ to modify the output in a richer, higher-dimensional subspace.

This directly motivates our proposed $\mathcal{L}_{\text{div}}$: by encouraging orthogonality among the selected vectors, we actively maximize their span, thereby enhancing the expressive capacity of the prompt-based bias.

# F  DISCUSSIONS

## F.1  EFFICIENCY ANALYSIS

We benchmark S-CMPT's training efficiency against the baselines (VPT and VFPT), the results are in Table 11:

Table 11: Efficiency benchmark of S-CMPT, VPT, and VFPT.

|  | GPU Time (Train) | GPU Memory (Train) | FLOPs (Train) |
|---|---|---|---|
| S-CMPT | 429.840 ms | 5.57 GB | 26.90 GFLOPs |
| VPT | 428.203 ms | 5.51 GB | 26.62 GFLOPs |
| VFPT | 433.391 ms | 5.51 GB | 26.62 GFLOPs |

All benchmarks are run with a batch size of 64 and 20 prompt vectors. The GPU Time is averaged over 100 steps after 10 warm-up steps. When calculating FLOPs, the batch size is set to 1.

The results show that the overhead from our selector and projector is marginal and does not create a computational bottleneck:

1. **Training Overhead is Marginal:** Our attention-based selector and projector indeed add computational cost. However, the added costs are marginal: +0.06 GB training memory and +0.28G FLOPs compared to the baselines.

2. **S-CMPT is Faster than the SOTA Baseline:** Importantly, S-CMPT's training time is faster than the SOTA baseline VFPT (which incurs costs for Fourier transforms) and is statistically negligible (+1.6 ms) compared to the simpler VPT.

This confirms our design choice: we accept a minor, one-time training overhead to enable our selection mechanism. This trade-off unlocks S-CMPT's primary efficiency gains at inference time: better performance with fewer prompt vectors.

## F.2  STATISTICAL SIGNIFICANCE OF FIGURE 2

The core idea of Figure 2 is that even the prompt vectors are selected randomly (random select), it can outperform CMPT who sticks to the "layer-wise correspondence" and "indiscriminate transfer" assumptions. To verify that our observations are statistically significant, we use a one-tailed binomial sign test to evaluate whether our "random selection" method (defined as a "success") is a statistically significant improvement over the baseline CMPT:

- Null Hypothesis: The random selection method is not better than CMPT. The probability of it outperforming CMPT in any given trial is at most 50% (i.e., $p_i$=0.5).

- Number of Trials (n): 36

- Number of successes (k): 34 (trials where random selection outperformed CMPT)

- Probability of success under the Null Hypothesis (p): 0.5

Using these parameters (as confirmed with `scipy.stats.binomtest`), we calculate the p-value:

```
>>> from scipy.stats import binomtest
>>> result = binomtest(k=34, n=36, p=0.5)
>>> print(result.pvalue)
>>> 1.941225491464138e-08
```

The p-value, or the probability of observing 34 successes in 36 trials if the null hypothesis were true, is $1.94 \times 10^{-8}$. This p-value is vanishingly small and falls far below any standard significance level. We therefore strongly reject the null hypothesis.

### F.3 LIMITATION ANALYSIS

To better understand the boundary of S-CMPT, here we provide a deep analysis on cases where S-CMPT provides marginal gains over the baseline. We have identified 2 distinct scenarios where the gains are limited, which can in turn validate our paper's core hypotheses.

**Limitation 1: Limited Improvements on 'Non-Distant' Tasks**   As our paper notes, S-CMPT's gains are are smaller on 'non-distant' tasks. We surpass the seminal VPT by +2.31% on 'non-distant' tasks, which is a smaller margin than the +5.96% gain on 'distant' tasks. This is an expected boundary. Our method's primary benefit is 'injecting filtered, task-relevant linguistic knowledge' to bridge significant semantic gaps. On 'non-distant' tasks, the ViT backbone's pretrained visual features are already highly effective and well-aligned with the task. There is less of a semantic gap to bridge, so the additional linguistic knowledge provides a smaller, though still positive, benefit.

**Limitation 2: Limited Improvements on MoCo v3 Backbone**   Our gains are smallest with the MoCo v3 backbone (e.g., +0.18% over CMPT, +1.34% over VFPT ). To analyze this, we looked at the Linear Probing performance (Jia et al., 2022) of the backbones, which reflects their "out-of-the-box" feature quality (Table 12). Marginal improvements on MoCo v3 backbone cannot be seen as a

Table 12: The linear probing performance of the three ViT backbones.

|  | VTAB-Natural | VTAB-Specialized | VTAB-Structured | Overall |
|---|---|---|---|---|
| ViT-SUP | 68.93 | 77.16 | 26.84 | 52.94 |
| ViT-MAE | 18.87 | 53.72 | 23.70 | 28.24 |
| ViT-MoCo v3 | 67.46 | 81.08 | 30.33 | 54.69 |

failure of S-CMPT, but rather a "ceiling effect" imposed by the backbone's strength:

- The MoCo v3 backbone is exceptionally strong, achieving the highest overall Linear Probing score (54.69). Because its baseline features are already so semantically rich, the "headroom" to add value is smaller.

- This contrasts sharply with the MAE backbone, a reconstructive method with far weaker linear probe features (28.24). This weak baseline has massive headroom for improvement, which is why S-CMPT's selective knowledge injection provides an enormous +9.60% gain over the SOTA VFPT baseline.

S-CMPT is still the top-performing method on MoCo v3, but its relative gain is bounded by the high quality of the underlying features.

