# OpenReview forum: "Autonomous Knowledge Integration Enables Efficient Cross-Modality Prompt Transfer"
_ICLR.cc/2026/Conference — Submitted to ICLR 2026_

### Official Review · Reviewer_Pynv · 2025-10-30

**Soundness:** 3
**Presentation:** 3
**Contribution:** 2
**Rating:** 4
**Confidence:** 4

**Summary:**

This paper addresses limitations in Cross-Modality Prompt Transfer (CMPT), which transfers prompts pretrained on data-rich modalities (e.g., text) to improve prompt tuning for data-scarce tasks in other modalities (e.g., vision). The authors identify two key problems with existing CMPT approaches: (1) previous work has been limited to shallow architectures, and (2) conventional methods transfer all source prompt vectors indiscriminately without accounting for redundancy or incompatibilities between source and target tasks.

To address these issues, the authors propose Selective Cross-Modality Prompt Transfer (S-CMPT), which introduces an attention-based prompt selector that identifies the most relevant source vectors for each target layer, along with a diversity regularization term to encourage selection of non-redundant vectors. The selected vectors are then adapted via linear projection to the target task. The method is evaluated on the VTAB-1k benchmark across multiple ViT backbones (supervised ImageNet-21k, MAE, and MoCo v3 pretraining), demonstrating significant performance improvements over existing methods while using fewer prompt vectors (20 per layer vs. 49.3 for the state-of-the-art VFPT method).

**Strengths:**

1. The paper identifies an important limitation in existing CMPT approaches - the assumption of layer-wise correspondence and indiscriminate transfer of all prompt vectors. The random selection experiment (Figure 2) compellingly demonstrates that layer-wise correspondence is suboptimal, with random selection outperforming CMPT in 94.4% of trials.

2. S-CMPT introduces a conceptually straightforward approach (attention-based selection + diversity regularization) that significantly outperforms existing methods. The elegance of the solution is a strength, avoiding unnecessary complexity while addressing a genuine problem.

3. The paper provides thorough evaluation across multiple pretraining settings (supervised, MAE, MoCo v3) and VTAB-1k tasks, with detailed per-task results. The ablation studies on selection range and diversity regularization strength provide convincing evidence for the design choices.

4. By using only 20 prompt vectors per layer (compared to 49.3 for VFPT), S-CMPT achieves better performance with significantly fewer parameters, which is crucial for parameter-efficient transfer learning.

**Weaknesses:**

1. While the empirical results are strong, the paper lacks theoretical justification for why certain prompt vectors are more transferable across modalities. A deeper analysis of what properties make vectors transferable would strengthen the contribution.

2. The paper shows that diversity regularization helps but doesn't thoroughly analyze why the specific cosine similarity-based approach works better than alternatives, or provide deeper insights into how diversity affects transfer performance.

3. The paper focuses on positive results but doesn't thoroughly analyze scenarios where S-CMPT provides minimal gains or underperforms baselines. Understanding the boundaries of the method's effectiveness would strengthen the contribution.

4. The paper claims to be the first method for deep architectures in CMPT, but doesn't thoroughly establish why previous CMPT work was limited to shallow architectures or why this transition is non-trivial.

5. The paper mentions pretraining the source prompt on SNLI with RoBERTa but lacks sufficient details about the pretraining process (e.g., number of epochs, convergence behavior, specific hyperparameters beyond what's in the supplement).

6. The technical novelty is limitted in a simple prompt selection module and a diversity regularization scheme. The performance gains are also marginal in most cases.

**Questions:**

- The statistical significance of the random selection experiment (Figure 2) isn't quantified. While 34 out of 36 trials showing improvement is compelling, a formal statistical test would strengthen this evidence.

- The paper doesn't explore how the method performs with different source modalities beyond text (SNLI/RoBERTa). Testing with other source modalities would strengthen claims about cross-modality generality.

- The paper could have included computational efficiency metrics (training/inference time) alongside parameter efficiency analysis to provide a more complete picture of practical benefits.

---

> ### Author Response · Authors · 2025-11-19
> **Response to weaknesses 1 & 2**
>
> ### **Weakness 1: While the empirical results are strong, the paper lacks theoretical justification for why certain prompt vectors are more transferable across modalities. A deeper analysis of what properties make vectors transferable would strengthen the contribution**
>
> We thank the reviewer for this fundamental question.
> Based on our experiments and understanding,
> "transferable" prompt vectors should have the following property:
> **their rank (or span in the feature space) should be big**.
>
> The logic behind this is simple:
> Theoretically,
> prompt vectors that are more spanned in the feature space can offer greater expressiveness,
> thereby offering greater transfer performance.
> As the reviewer notes, this is closely tied to our diversity regularization, and we will provide the full theoretical justification for this in our response to Weakness 2.
>
> ### **Weakness 2. The paper shows that diversity regularization helps but doesn't thoroughly analyze why the specific cosine similarity-based approach works better than alternatives, or provide deeper insights into how diversity affects transfer performance.**
>
> Our choice of cosine similarity is a direct, formal, and theoretically justified proxy for maximizing the "span" of the selected prompt vectors.
>
> Petrov et al. [1] had shown:
> the function of prompts is **adding a bias term to an attention block's output**,
> where this bias is limited to **the span (or rank) of the prompt vectors**.
>
> Let's consider an extreme case:
> a selector chooses collinear or highly similar prompt vectors.
> This would apparently cause a rank collapse,
> restricting the prompt's influence to the outputs to a single, unexpressive direction.
>
> Our Ldiv directly optimizes this by minimizing cosine similarity,
> which pushes the vectors toward orthogonality.
> This maximizes the span and effective rank of the selected prompt subspace,
> thereby maximizing the expressiveness of the selected prompt vectors and boosting transfer performance.
>
> Therefore, Ldiv is not an arbitrary choice;
> it is a principled mechanism to ensure the prompts can bias the model's representations in a rich,
> multi-dimensional way.
> We chose cosine similarity as it is the most standard and direct way to formalize this 'span maximization' objective.
>
> This theoretical basis has been updated in our supplementary.
>
> [1] Petrov, Aleksandar, Philip Torr, and Adel Bibi. "When Do Prompting and Prefix-Tuning Work? A Theory of Capabilities and Limitations." In The Twelfth International Conference on Learning Representations.

---

> ### Author Response · Authors · 2025-11-19
> **Response to weakness 3**
>
> ### **Weakness 3. The paper focuses on positive results but doesn't thoroughly analyze scenarios where S-CMPT provides minimal gains or underperforms baselines. Understanding the boundaries of the method's effectiveness would strengthen the contribution.**
>
> We thank the reviewer for this excellent point.
> Understanding the method's boundaries is essential.
> We have identified 2 distinct scenarios where the gains are limited,
> which in turn validate our paper's core hypotheses.
>
> **Limitation 1: Limited improvements on "non-distant" tasks**
>
> As our paper notes,
> S-CMPT's gains are smaller on "non-distant" tasks.
> We surpass the seminal VPT by +2.31% on non-distant tasks,
> which is a smaller margin than the +5.96% gain on "distant" tasks.
>
> This is an expected boundary.
> Our method's primary benefit is "injecting filtered, task-relevant linguistic knowledge" to bridge significant semantic gaps.
> On "non-distant" tasks,
> the ViT backbone's pretrained visual features are already highly effective and well-aligned with the task.
> There is less of a semantic gap to bridge,
> so the additional linguistic knowledge provides a smaller, though still positive, benefit.
>
> **Limitation 2: Limited improvements on MoCo V3 backbone**
>
> Our gains are smallest with the MoCo v3 backbone (e.g., +0.18% over CMPT, +1.34% over VFPT ).
> To analyze this,
> we looked at the **Linear Probing** performance [1] of the backbones,
> which reflects their "out-of-the-box" feature quality.
>
> |                | Natural | Specialized | Structured | Overall |
> |----------------|---------|-------------|------------|---------|
> | ViT-Supervised | 68.93   | 77.16       | 26.84      | 52.94   |
> | ViT-MAE        | 18.87   | 53.72       | 23.70      | 28.24   |
> | ViT-MoCo v3    | 67.46   | 81.08       | 30.33      | 54.69   |
>
> This is not a failure of S-CMPT, but rather a "ceiling effect" imposed by the backbone's strength:
> + The MoCo v3 backbone is exceptionally strong,
> achieving the highest overall Linear Probing score (54.69).
> Because its baseline features are already so semantically rich,
> the "headroom" to add value is smaller.
> + This contrasts sharply with the MAE backbone,
> a reconstructive method with far weaker linear probe features (28.24).
> This weak baseline has massive headroom for improvement,
> which is why S-CMPT's selective knowledge injection provides an enormous +9.60% gain over the SOTA VFPT baseline.
>
> S-CMPT is still the top-performing method on MoCo v3,
> but its relative gain is bounded by the high quality of the underlying features.
>
> This limitation analysis has been added to the discussion section in our supplementary material.
>
> [1] Jia, Menglin, Luming Tang, Bor-Chun Chen, Claire Cardie, Serge Belongie, Bharath Hariharan, and Ser-Nam Lim. "Visual prompt tuning." In European conference on computer vision, pp. 709-727. Cham: Springer Nature Switzerland, 2022.

---

> ### Author Response · Authors · 2025-11-19
> **Response to weaknesses 4, 5, 6**
>
> ### **Weakness 4. The paper claims to be the first method for deep architectures in CMPT, but doesn't thoroughly establish why previous CMPT work was limited to shallow architectures or why this transition is non-trivial.**
>
> We thank the reviewer for this question.
> In our opinion,
> prior work [1] was limited to the shallow architecture because it was foundational and exploratory.
> It faces a brand-new problem.
> The best practice would be starting with the basic architecture (shallow prompt) to verify the feasibility.
>
> The transition from shallow to deep architectures is "non-trivial" because it exposes a critical,
> flawed assumption in the baseline CMPT model:
> + When scaling to deep (e.g., 12-layer) models,
> the "Hindrance" arises from the "layer-wise correspondence" assumption:
> should we keep it or break it?
> The best practice shall be explored empirically.
>
> We conducted experiments to test both scenarios and found
> that the assumption should not be kept and selective prompt transfer is superior.
> Our work is the first to formally identify this flaw,
> prove it empirically, and propose a principled, learned solution.
>
>
> [1] Zhang, Ningyuan, Jie Lu, Keqiuyin Li, Zhen Fang, and Guangquan Zhang. "Release the Powers of Prompt Tuning: Cross-Modality Prompt Transfer." In The Thirteenth International Conference on Learning Representations.
>
>
> ### **Weakness 5. The paper mentions pretraining the source prompt on SNLI with RoBERTa but lacks sufficient details about the pretraining process (e.g., number of epochs, convergence behavior, specific hyperparameters beyond what's in the supplement).**
>
> We thank the reviewer for pointing this out, as these details are important for reproducibility.
> We have added a detailed subsection to the appendix covering the source prompt pretraining.
> Below is a quick take-away:
>
> The source prompt (with RoBERTa-base) was trained on SNLI using the following settings:
>
> | Epoches | LR    | LR Decay? | LR Warm-Up? | WD | Bach Size |
> |---------|-------|-----------|-------------|----|-----------|
> | 64      | 0.001 | No        | 0           | 0  | 16        |
>
> The model converged very quickly and remained stable throughout training,
> as shown by the test accuracy on SNLI:
>
> | Epoch | 16    | 32    | 48    | 64    |
> |-------|-------|-------|-------|-------|
> | Acc   | 90.12 | 90.23 | 90.07 | 90.57 |
>
> As the table shows,
> performance was already stable at 90.12% by epoch 16 and remained consistent (±0.5%) through epoch 64.
> This indicates the pretraining was stable and had fully converged,
> providing a robust and consistent set of source vectors for our transfer experiments.
>
> ### **Weakness 6. The technical novelty is limitted in a simple prompt selection module and a diversity regularization scheme. The performance gains are also marginal in most cases.**
>
> Thank you for the thoughtful feedback.
> We respectfully disagree and would like to clarify both the novelty and effectiveness of S-CMPT.
>
> **Novelty**
>
> The two components in S-CMPT (the prompt selector and the diversity regularizer) are intentionally simple,
> but they directly and effectively address the fundamental limitations of existing CMPT methods:
> + Prompt Selector:
> Prior CMPT methods rely on a rigid layer-wise correspondence and transfer all source prompt vectors by default.
> Our analysis shows that these assumptions break down when scaling CMPT to deeper architectures.
> The selector is the first mechanism designed to replace the flawed assumptions with a selective, flexible, and layer-agnostic reuse of source prompt vectors.
> + Diversity regularization:
> Without additional constraints, the selector tends to collapse onto a small number of vectors.
> The diversity loss is introduced to ensure the selected prompts span a richer subspace,
> improving their information carriage and mitigating representational collapse.
>
> Thus, the contributions are not arbitrary or incremental;
> they form a minimal and principled solution to facilitate the transition from shallow to deep architecture.
> We view the insight: the baseline assumptions are flawed, as the core novelty.
> The fact that this insight leads to a lightweight,
> parameter-efficient solution is a strength, not a limitation,
> and aligns with the philosophy of PETL research.
>
> **Performance**:
>
> We would also like to clarify that the performance gains are strong and meaningful,
> especially in the context of parameter-efficient transfer learning.
> S-CMPT achieves +0.65% (Supervised), +9.60% (MAE), and +1.34% (MoCo v3) gains over the SOTA baseline VFPT
> while using 2.5× fewer prompt vectors.
> Such improvements,
> achieved with substantially reduced parameter count,
> represent a superior parameter–performance trade-off,
> which is the central objective of PETL.

---

> ### Author Response · Authors · 2025-11-19
> **Response to question 1**
>
> ### **Question 1. The statistical significance of the random selection experiment (Figure 2) isn't quantified. While 34 out of 36 trials showing improvement is compelling, a formal statistical test would strengthen this evidence.**
> This is an excellent point. We have performed a formal statistical test to quantify this observation.
> We use a one-tailed binomial sign test to evaluate whether our "random selection" method (defined as a "success") is a statistically significant improvement over the baseline CMPT:
> + Null Hypothesis:
> The random selection method is not better than CMPT.
> The probability of it outperforming CMPT in any given trial is at most 50% (i.e., p<=0.5).
> + Number of Trials (n): 36
> + Number of successes (k): 34 (trials where random selection outperformed CMPT)
> + Probability of success under the Null Hypothesis (p): 0.5
>
> Using these parameters (as confirmed with `scipy.stats.binomtest`), we calculated the p-value:
>
> ```
> from scipy.stats import binomtest
> result = binomtest(k=34, n=36, p=0.5)
> print(result.pvalue)
> ```
>
> The p-value,
> or the probability of observing 34 successes in 36 trials if the null hypothesis were true,
> is 1.94e-08.
> This p-value is vanishingly small and falls far below any standard significance level.
> We therefore strongly reject the null hypothesis.
>
> The statistical analysis has been added to the discussion section in our supplementary.

---

> ### Author Response · Authors · 2025-11-19
> **Response to question 2**
>
> ### **Question 2. The paper doesn't explore how the method performs with different source modalities beyond text (SNLI/RoBERTa). Testing with other source modalities would strengthen claims about cross-modality generality.**
>
> We thank the reviewer for the valuable suggestion.
> Indeed, exploring more source modalities can strengthen our generalization ability.
>
> For now, our choice of text as the source modality was a deliberate one,
> designed to test the most challenging and practical scenario for our paper's core hypothesis.
> We had two primary reasons for this:
> 1. It Represents the Core Problem:
> Our study aims to leverage a data-rich modality to address data scarcity in a target task.
> Text is the most abundant and resource-rich modality,
> making it the ideal candidate for this problem.
> 2. It Provides the Hardest Test:
> The transfer from text to vision creates a significant modality gap.
> Successfully proving our method in this scenario provides the most convincing evidence for our claims.
> 3. To keep in-line with prior works:
> CMPT was initially experimented on language-to-vision tasks.
> To directly build upon and fairly compare our work to the method we are improving,
> we adopted the same established language-to-vision modality pairing.
>
> While we did not test other source modalities,
> our "select-then-transfer" framework is,
> by design,
> modality-agnostic.
> The selector mechanism operates on a global pool of source vectors,
> and it would function identically on prompts pretrained on any modality.
>
> We agree that exploring other source modalities is a promising direction, we will add this to our future work.

---

> ### Author Response · Authors · 2025-11-19
> **Response to question 3**
>
> ### **Question 3. The paper could have included computational efficiency metrics (training/inference time) alongside parameter efficiency analysis to provide a more complete picture of practical benefits.**
>
> We have benchmarked S-CMPT's efficiency against the baselines:
>
> |        | GPU Time (Train) | GPU Memory (Train) | FLOPs (Train) |
> |--------|------------------|--------------------|---------------|
> | S-CMPT | 429.840 ms       | 5.57 GB            | 26.90 GFLOPs  |
> | VPT    | 428.203 ms       | 5.51 GB            | 26.62 GFLOPs  |
> | VFPT   | 433.391 ms       | 5.51 GB            | 26.62 GFLOPs  |
>
> _All benchmarks run with a batch size of 64 and Np=20.
> Time is averaged over 100 steps.
> FLOPs are for batch size 1.
> The script is `efficiency_comparison.py` in the newly updated .zip file_
>
> The results show that the overhead from our selector and projector is marginal and does not create a computational bottleneck:
> 1. Training Overhead is Marginal:
> Our attention-based selector and projector indeed add computational cost.
> However, the added costs are marginal: +0.06 GB training memory and +0.28G FLOPs compared to the baselines.
> 2. S-CMPT is Faster than the SOTA Baseline:
> Importantly, S-CMPT's training time is faster than the SOTA baseline VFPT
> (which incurs costs for Fourier transforms) and is statistically negligible (+1.6 ms) compared to the simpler VPT.
>
> This confirms our design choice:
> we accept a minor, one-time training overhead to enable our selection mechanism.
> This trade-off unlocks S-CMPT's primary efficiency gains at inference time.
>
> The selector and projector "are discarded" after training.
> S-CMPT's inference cost is therefore determined only by its final, compact prompt length (Np=20).
> This makes our method ~2.5x more parameter-efficient (20 vs. 49.3 vectors) and correspondingly faster at inference (fewer vectors to process) than VFPT,
> all while achieving superior accuracy.
>
> The computational efficiency comparison has been added to the discussion section in our supplementary material.

---

> > ### Comment · Reviewer_Pynv · 2025-11-24
> >
> > Thanks for the rebuttal. I still have the following concerns about this paper: 1) The relation between the prompt diversity and feature expressiveness or transfer performance  is a little obscure. 2) The performance gain over the baseline method CMPT seems marginal in most cases.

---

> > > ### Author Response · Authors · 2025-11-25
> > > **Response to Concern 1**
> > >
> > > We sincerely thank you for your continued engagement.
> > > We understand your remaining concerns and would like to address them directly with evidence from our new experiments.
> > >
> > > ### **Response to Concern 1: The relation between prompt diversity and transfer performance**
> > >
> > > We apologize if this relationship appeared obscure.
> > > Here we demonstrate that diversity is not merely a heuristic,
> > > but a necessary condition for feature expressiveness.
> > > We support this claim with theoretical grounding and two specific case studies.
> > >
> > >
> > > ### **1. Theoretical Mechanism**
> > > As detailed in our new supplementary section,
> > > prompt vectors function as an additive bias term in the attention block.
> > > The expressiveness of this bias is mathematically bounded by the effective rank (or span) of the selected prompt vectors.
> > > If selected vectors exhibit low diversity (high cosine similarity), they'd become more collinear.
> > > This causes a "rank collapse," restricting the bias term to narrow directions.
> > > Maximizing diversity promotes the orthogonality of vectors,
> > > maximizing the span and ensuring the bias term can modify representations in a rich, multi-dimensional subspace.
> > > **The above theory had been verified by a published work** [1].
> > >
> > > ### **2. Empirical Support**
> > >
> > > In the following case studies, we show:
> > > 1. Low diversity leads to a rank collapse in the selected vectors, and a consequent drop in transfer performance.
> > > 2. Transfer performance is highly correlated with vector diversity, instead of source task performance.
> > >
> > > ### **2.1 Case Study A**
> > > To support the theoretical mechanism,
> > > we conducted a '_harmful selection_' case study on CIFAR
> > > where we force the selector to choose low-diversity vectors by setting $\lambda = -0.01$.
> > >
> > > | Metric              | High Diversity ($\lambda=0.01$) | Normal ($\lambda=0$) | Low Diversity ($\lambda=-0.01$) |                         |
> > > |---------------------|---------------------------------|----------------------|---------------------------------|-------------------------|
> > > | Cosine Similarity   | 0.0446                          | 0.0708               | 0.5360                          | Lower: High diversity   |
> > > | Selection Entropy   | 4.1741                          | 4.2943               | 2.4800                          | Higher: High diversity  |
> > > | Max Attention       | 2.2496                          | 2.3368               | 10.5205                         | Lower: High diversity   |
> > > | Accuracy            | 82.86                           | 81.76                | 81.52                           |                         |
> > >
> > > Here's how each metric is calculated:
> > > + Selection Entropy: The avg. entropy of the aggregated softmax attention.
> > > + Max Attention: The avg. max value of the aggregated softmax attention.
> > > + \# Vectors (A > 0.5): The avg. count of source vectors with significant attention (>0.5).
> > > + Cosine Similarity: The avg. cosine similarity among the selected prompt vectors.
> > >
> > > This experiment isolates the variable.
> > > Forcing low diversity (High Cosine Sim: 0.536) directly causes a rank collapse (low entropy, spiky attention)
> > > and a consequent drop in transfer performance (81.52%).
> > > This confirms that diversity is the active mechanism preserving performance.
> > >
> > > ### **2.2 Case Study B**
> > > To show diversity is a general property of transferability,
> > > we transfer source prompts at different source pretraining epoches using standard CMPT (no selection, no diversity regularization, only brutal-force transfer).
> > > The results are averaged on 10 seeds (just to rule out the influence of randomness)：
> > >
> > > | Epoch | SNLI Acc. | Cos.  | CIFAR Acc.  |
> > > |-------|-----------|-------|-------------|
> > > | 1     | 86.89     | 0.478 | 81.25±0.46  |
> > > | 37    | 90.94     | 0.147 | 81.61±0.38  |
> > > | 64    | 90.57     | 0.129 | 81.67±0.42  |
> > >
> > > The results show that transferability is strongly correlated with vector diversity, rather than source accuracy.
> > >
> > > Both case studies and the theoretical bound confirm that diversity is the key metric that prevents rank collapse and enables effective transfer.
> > > We sincerely hope they could solve your concern, thank you!
> > >
> > > [1] Petrov, Aleksandar, Philip Torr, and Adel Bibi. "When Do Prompting and Prefix-Tuning Work? A Theory of Capabilities and Limitations." In The Twelfth International Conference on Learning Representations.

---

> > > ### Author Response · Authors · 2025-11-25
> > > **Response to Concern 2**
> > >
> > > ### **Response to Concern 2: Marginal performance gain over CMPT**
> > >
> > > We acknowledge that the average gain over our re-implemented CMPT-Deep is modest in settings with strong backbones (+0.38% on Supervised, +0.18% on MoCo v3).
> > > However, we argue that S-CMPT provides critical value beyond these averages, demonstrated by magnitude in difficult settings and statistical significance.
> > > 1. The gains are not modest when 'headroom' is large:
> > > on MAE where the backbone features are less mature (low linear probing performance),
> > > S-CMPT outperforms CMPT by +2.44% and the SOTA VFPT by +9.60%.
> > > This demonstrates that when the backbone actually needs external knowledge,
> > > S-CMPT's selective mechanism is crucial,
> > > whereas the rigid CMPT struggles.
> > > 2. Even if margins are modest, S-CMPT is statistically superior.
> > > We analyzed the consistency of improvement across all 57 experimental trials (19 tasks $\times$ 3 backbones) reported in our tables.
> > >     + Null Hypothesis: S-CMPT is not better than CMPT. The probability of S-CMPT outperforming CMPT in any given trial is at most 50% (i.e., p<=0.5).
> > >     + S-CMPT outperforms CMPT in 38 out of 57 trials.
> > >     + The calculated p value is 0.016.
> > >
> > > Since $p < 0.05$, we reject the null hypothesis.
> > > This confirms that S-CMPT is statistically significantly better than the baseline.
> > > The modules are consistently "doing their job," providing robustness even when the numerical ceiling is high.
> > >
> > > Finally, it is important to note that CMPT-Deep is not a weak, pre-existing baseline;
> > > it is a strong baseline **we implemented ourselves** to test deep transfer.
> > > Beating it consistently is a significant result.
> > >
> > > We intentionally kept S-CMPT in its **minimal form** (simple attention + regularization) to isolate our core contribution:
> > > **breaking the "rigid layer-wise correspondence" assumption**.
> > > Our goal was to prove that breaking this assumption is the key,
> > > rather than achieving gains through complex, over-engineered designs.
> > > We are confident that future work can build more complex architectures on top of this validated "select-then-transfer" paradigm to achieve even larger margins.

---

### Official Review · Reviewer_XTPd · 2025-10-30

**Soundness:** 3
**Presentation:** 2
**Contribution:** 3
**Rating:** 4
**Confidence:** 4

**Summary:**

This paper proposes a new method called Selective Cross Modal Hint Transfer (S-CMPT), aimed at optimizing cross modal knowledge transfer by automatically selecting the most relevant source hint vectors. This method has its novel theoretical assumptions (challenge layer correspondence and validation source hint redundancy), excellent performance and parameter efficiency, and clear structure.

**Strengths:**

The core contribution of the article lies in clearly proposing and verifying two key assumptions in cross modal prompt transfer: redundancy in source prompts and suboptimal strict inter layer correspondence. This insight, challenges traditional beliefs through empirical analysis and emphasizes the superiority of selective knowledge adoption over indiscriminate reuse. The proposed method maintains excellent efficiency-efficacy trade-off compared with other SOTA methods.

**Weaknesses:**

Well-structured paper providing organized motivation but not enough innovation. One motivation of the author is that the original CMPT only used prompts from the corresponding layer, which is redundant and incompatible. Therefore, the selection range of prompts was extended from the corresponding layer to all layers; Although the results may seem effective, they are essentially an improvement in the selection range or hyperparameter configuration, without innovation in the logic of motivation or deeper theoretical exploration.


One of the motivations for the proposed "diversity regularization" is to "promote the use of different vectors by forcing more diverse attention distributions, thereby helping to obtain complementary information for the target task". Although this motivation is reasonable, it lacks specific examples or theoretical basis on how to measure or formalize "complementary information" in existing work, resulting in a lack of rigor in the contribution of this theory.

The performance improvement seems limited.

**Questions:**

The description in Figure 5 is confusing as to which two parts each row and column in the attention diagram are specifically calculated, and why there is a subscript j for Q in Eq. 5 but not in Eq. 2. How many Q are there in each layer, and what are the dimensions of each Q.

---

> ### Author Response · Authors · 2025-11-19
> **Response to weaknesses 1 & 2**
>
> ### **Weakness 1: Well-structured paper providing organized motivation but not enough innovation. One motivation of the author is that the original CMPT only used prompts from the corresponding layer, which is redundant and incompatible. Therefore, the selection range of prompts was extended from the corresponding layer to all layers; Although the results may seem effective, they are essentially an improvement in the selection range or hyperparameter configuration, without innovation in the logic of motivation or deeper theoretical exploration.**
>
> Thank you for your credits!
> About our work being an improvement in hyperparameter configuration and without innovation in the logic of motivation or deeper theoretical exploration,
> we respectfully disagree.
> Our work's primary innovation is to challenge the baseline's core assumption of rigid layer-wise correspondence.
> We are the first to hypothesize that this assumption is suboptimal for deep architectures.
>
> Our contribution is not just 'extending the range' or 'hyperparameter configuration', but:
> 1. Empirically Proving the Flaw:
> Our pilot experiment (Figure 2) provides the "deeper exploration" the reviewer is asking for.
> It shows that even random layer-agnostic selection outperforms the rigid baseline in 94.4% of trials.
> This proves the baseline's 1-to-1 logic is flawed.
> 2. Proposing a Principled Solution:
> S-CMPT is the first method to introduce a "flexible, layer-agnostic reuse of source prompt vectors".
> Our ablation study (Figure 4) further validates this,
> showing that our "ALL" (all-layer) selection "surpasses both ADJACENT and LOCAL in 9 out of 12 cases".
>
> This is a novel conceptual contribution,
> not a hyperparameter tweak,
> and it is validated by both our pilot study and main results.
>
> About the theoretical exploration, let's move on to weakness 2.
>
> ### **Weakness 2: One of the motivations for the proposed "diversity regularization" is to "promote the use of different vectors by forcing more diverse attention distributions, thereby helping to obtain complementary information for the target task". Although this motivation is reasonable, it lacks specific examples or theoretical basis on how to measure or formalize "complementary information" in existing work, resulting in a lack of rigor in the contribution of this theory.**
>
> We thank the reviewer for this crucial point, which allows us to add a deeper theoretical basis for our method.
>
> "Complementary information" is formally equivalent to the effective rank or span of the prompt vectors.
> We measure "complementary information" using pairwise cosine similarity as a direct proxy,
> which is a reflection of the angles among prompt vectors.
> Minimizing similarity is a standard method for encouraging orthogonality and maximizing rank,
> which motivates us to conceptualize our diversity regularization.
> This is supported by the following **theoretical basis** (which has also been added to our supplementary material):
>
> + A recent theoretical analysis (Petrov et al., [1]) has shown that adding a sequence of prompt vectors to an attention block is equivalent to adding a bias term to its output.
> This bias term's expressiveness is limited to a subspace with a rank no greater than the length of the prompt sequence.
> + Consider an extreme case where all selected prompt vectors are collinear (i.e., they are experiencing rank collapse and have a rank of 1).
> This is the exact "high redundancy" scenario.
> In this case, no matter how the model learns to weigh them,
> the output bias term is restricted to a single direction.
>
> This brings us to our contribution:
> + Our Ldiv is the explicit mechanism to prevent this rank collapse.
> By minimizing cosine similarity,
> we push the vectors towards orthogonality,
> thereby maximizing the span and effective rank of the selected prompt subspace.
> + Our Ldiv is the first mechanism proposed in this context to explicitly optimize for this property,
> ensuring the prompts can bias the model's representations in a rich,
> multi-dimensional way,
> rather than a single, collapsed direction.
>
> This theoretical basis has been added to our supplementary material.
> We sincerely hope this can resolve your concern. Thank you!
>
> [1] Petrov, Aleksandar, Philip Torr, and Adel Bibi. "When Do Prompting and Prefix-Tuning Work? A Theory of Capabilities and Limitations." In The Twelfth International Conference on Learning Representations.

---

> ### Author Response · Authors · 2025-11-19
> **Response to weakness 3**
>
> ### **Weakness 3: The performance improvement seems limited.**
>
> We respectfully disagree with the characterization of our improvements as 'marginal.'
>
> 1. Gains over SOTA (VFPT):
> In the highly-optimized VTAB-1k benchmark,
> S-CMPT consistently outperforms the SOTA.
> It achieves a +0.65% gain (Supervised ViT),
> a +9.60% gain (MAE ViT),
> and a +1.34% gain (MoCo v3) over VFPT .
> A +0.65% gain over a 2024 SOTA baseline is significant,
> and the ~1-10% gains on self-supervised models are substantial.
>
> 2. The Core Claim (Efficiency-Efficacy):
> The paper's main contribution is not accuracy alone,
> but the parameter-performance trade-off.
> S-CMPT achieves these SOTA results while using only 20 vectors per layer, which is 2.5x fewer prompt vectors than the SOTA VFPT (49.3 vectors) and 3x fewer than VPT (60.9 vectors).
>
> Achieving better performance with 2.5x fewer parameters is the opposite of a marginal improvement; it demonstrates a new, superior trade-off for efficient transfer learning.

---

> ### Author Response · Authors · 2025-11-19
> **Response to questions**
>
> ### **The description in Figure 5 is confusing as to which two parts each row and column in the attention diagram are specifically calculated, and why there is a subscript j for Q in Eq. 5 but not in Eq. 2. How many Q are there in each layer, and what are the dimensions of each Q.**
>
> Thank you for pointing this out!
> Let's sort this out step by step:
> 1. How many Q are there:
> There is one learnable query matrix $Q$ per target layer. Since the ViT has 12 layers, there are 12 $Q$ matrices in total, denoted $Q^{(1)}, Q^{(2)}, ..., Q^{(12)}$.
> 2. What are the dimensions:
> The dimension of each $Q^{(i)}$ matrix is $N_p \times d_h$, or $20 \times 768$ in our experiments.
> $N_p=20$ is the number of queries (and thus the number of vectors we will select) for that layer, and $d_h=768$ is the hidden dimension of ViT.
> 3. Eq. 2 vs. Eq. 5:
>    + Eq. 2 ($...Q^{(i)}...$): This is the main operational equation for the forward pass.
>    Here, $Q^{(i)}$ refers to the entire $20 \times 768$ query matrix for layer i.
>    + Eq. 5 ($...Q_j^{(i)}...$): This equation is only for the visualization in Figure 5.
>    Here, $Q_j^{(i)}$ refers to the j-th row (a single $1 \times 768$ query vector) of the $Q^{(i)}$ matrix.
>    This equation describes how the "aggregated attention distribution" is calculated.
> 4. Figure 5 Rows/Columns:
>    + Columns:
>    The columns represent the entire source prompt pool $p_s$ (all $12 \times 20 = 240$ vectors),
>    grouped by their original source layer ($p_s^{(1)}$ to $p_s^{(12)}$).
>    + Rows:
>      1. How to get a single row? Take the first row for example.
>      2. Remember we have 20 query vectors for the first layer of ViT (each vector is a row of $Q^{(1)}$).
>      3. We take the first query vector and calculate the attention scores with the 240 source prompt vectors. The resultant attention scores would be a 1*240 vector.
>      4. Then we take the second query vector and calculate the attention scores.
>      5. Until we are done with all the 20 query vectors, we add all the attention scores up, the result is a 1*240 vector in the first row of Figure 5.
>
> We sincerely hope this could solve your concern. Thank you!

---

### Official Review · Reviewer_tRV6 · 2025-10-31

**Soundness:** 3
**Presentation:** 2
**Contribution:** 2
**Rating:** 4
**Confidence:** 4

**Summary:**

This paper addresses inefficiencies in Cross-Modality Prompt Transfer (CMPT), a technique for adapting pretrained models from data-rich modalities (like text) to data-scarce ones (like vision). The authors argue that existing methods suffer from indiscriminately transferring all source prompt vectors, which can include redundant or even detrimental information. This problem is compounded by a rigid layer-wise correspondence assumption, where vectors from a source layer are only transferred to the corresponding target layer, which the paper shows is suboptimal. To solve this, the authors propose Selective Cross-Modality Prompt Transfer (S-CMPT), a method designed for deep architectures that follows a "select-then-transfer" paradigm. S-CMPT uses a lightweight attention mechanism as a "prompt selector" to identify the most relevant vectors from the entire source prompt pool for each target layer. It also introduces a novel diversity regularization term to encourage the selection of diverse, non-redundant vectors. Experiments on the VTAB-1k benchmark demonstrate that S-CMPT achieves state-of-the-art accuracy, outperforming previous prompt-based methods while using significantly fewer parameters.

**Strengths:**

1. The paper's primary strength is its clear identification of a logical flaw in existing methods, vector redundancy and rigid layer-wise transfer and the proposal of an elegant, simple, and highly effective solution. The authors' hypothesis is convincingly validated before introducing their method via a pilot experiment, which demonstrates that even randomly selecting vectors outperforms the rigid layer-wise baseline.

2. The proposed S-CMPT method achieves an efficiency-efficacy trade-off by setting a new state-of-the-art on the VTAB-1k benchmark while using 2.5x fewer prompt vectors than the next-best method. This claim is thoroughly supported by extensive ablation studies that systematically test the key design choices, namely the vector selection range (Local, Adjacent, or All) and the impact of the diversity regularization term, proving the robustness of their approach.

3. Finally, the paper includes helpful visualizations that provide a clear intuition for why the diversity regularization is effective, showing how it forces the model to attend to a broader, less redundant set of source vectors

**Weaknesses:**

1. The related works are not fully discussed. Only a few works are incorporated and few state-of-the-art methods are included.
2. The paper has claimed that Prior work on CMPT has primarily focused on shallow architectures. However, the paper doesn't demonstrate how about the effects of shallow architectures? How does the proposed method overcome this?
3. The proposed method is relatively simple. A attention operation is adopted to select prompts from the source model, and a diversity loss is used to prompt source distribution.
4. The methods included for comparison are a bit old. Some state-of-the-art methods are not included. Only one work published in 2024 is included. More works should be discussed. For example:
[1] DePT: Decomposed Prompt Tuning for Parameter-Efficient Fine-tuning, CVPR 2024
[2] M2PT: Multimodal Prompt Tuning for Zero-shot Instruction Learning, EMNLP2024
[3] FedMVP: Federated Multi-modal Visual Prompt Tuning for Vision-Language Models, ICCV2025
[4] PromptKD: Unsupervised Prompt Distillation for Vision-Language Models, CVPR2024
5. The performance improvement is marginal on most settings. For example, only +0.38%,+2.44%,and +0.18% on VTAB-1k with MoCo V3 pretrained weights.

**Questions:**

See above

---

> ### Author Response · Authors · 2025-11-19
> **Response to weaknesses 1 & 4**
>
> ### **Weaknesses 1 & 4: Related Works**
>
> We sincerely thank the reviewer for highlighting this.
> We agree that the original Related Work section did not sufficiently reflect the breadth of the prompt-tuning literature.
> In the revised submission,
> we have substantially expanded this section and referenced the mentioned works.
> The related work section is now reorganized into three parts:
> language, vision, and vision–language prompt tuning, to provide a clearer and more comprehensive overview.
> The newly uploaded PDF reflects these updates.
>
> Regarding benchmark comparisons, we would like to clarify the following:
>
> 1. M2PT [1], FedMVP [2], and PromptKD [3] are designed for vision–language models (VLMs) that contain both a vision encoder and a large language encoder/decoder.
> These methods are evaluated in VLM-specific instruction-tuning or multimodal learning settings, and none of them are benchmarked on VTAB-1k.
> In contrast, our work focuses on single-vision-encoder architectures, and all baselines we compare against (VPT and VFPT) share this structural assumption and have been evaluated on VTAB-1k.
> Therefore, directly comparing S-CMPT with VLM-based methods would be structurally mismatched and less informative.
> 2. DePT [4] is more relevant because it is designed for single-model prompt tuning, originally for language models.
> With a bit of efforts modifying it, it can be adapted to ViT.
> To address this, we have re-implemented DePT for ViT and performed hyperparameter search following the dual-learning-rate strategy described in the original paper.
> Below we report the results on some representative VTAB-1k tasks:
>
> |        | Cifar    | Flowers  | Pets     | Resisc45 | DmLab    | Kitti    |
> |--------|----------|----------|----------|----------|----------|----------|
> | VFPT   | 80.7     | **99.3** | **90.3** | 84.4     | 48.3     | 79.3     |
> | DePT   | 81.6     | 98.4     | 89.3     | 84.0     | 45.1     | 78.9     |
> | S-CMPT | **82.7** | 98.6     | 90.1     | **85.7** | **48.5** | **80.3** |
>
> Our implementation of DePT is included in the supplementary zip file:
> + `./model/dept.py`: DePT module.
> + `./grid_search_dept_sup.py`: Hyperparameter grid search that supports dual learning rate adopted in their paper.
> + `./train_dept_sup.py`: Training script.
>
> Finally, we emphasize that every necessary prompt-tuning method designed for ViT and benchmarked on VTAB-1k has been included in our comparison.
> This ensures that our evaluation is fair, representative, and directly aligned with the setting studied in this paper.
>
> [1] Wang, Taowen, Yiyang Liu, James Chenhao Liang, Yiming Cui, Yuning Mao, Shaoliang Nie, Jiahao Liu et al. "M $^ 2$ PT: Multimodal Prompt Tuning for Zero-shot Instruction Learning." arXiv preprint arXiv:2409.15657 (2024).
>
> [2] Singha, Mainak, Subhankar Roy, Sarthak Mehrotra, Ankit Jha, Moloud Abdar, Biplab Banerjee, and Elisa Ricci. "FedMVP: Federated Multi-modal Visual Prompt Tuning for Vision-Language Models." arXiv preprint arXiv:2504.20860 (2025).
>
> [3] Li, Zheng, Xiang Li, Xinyi Fu, Xin Zhang, Weiqiang Wang, Shuo Chen, and Jian Yang. "Promptkd: Unsupervised prompt distillation for vision-language models." In Proceedings of the IEEE/CVF Conference on Computer Vision and Pattern Recognition, pp. 26617-26626. 2024.
>
> [4] Shi, Zhengxiang, and Aldo Lipani. "Dept: Decomposed prompt tuning for parameter-efficient fine-tuning." arXiv preprint arXiv:2309.05173 (2023).

---

> ### Author Response · Authors · 2025-11-19
> **Response to weaknesses 2, 3, 5**
>
> ### **Weakness 2: The paper has claimed that Prior work on CMPT has primarily focused on shallow architectures. However, the paper doesn't demonstrate how about the effects of shallow architectures? How does the proposed method overcome this?**
>
> Thank you for raising this point.
> We did not include shallow CMPT in our main comparison because
> **shallow CMPT performs substantially worse than deep CMPT.**
> As shown in the below table:
>
> |                  | VTAB-Natural | VTAB-Specialized | VTAB-Structured |
> |------------------|--------------|------------------|-----------------|
> | CMPT-Shallow | 80.55        | 84.12            | 53.47           |
> | CMPT-Deep        | 81.48        | 85.55            | 60.44           |
> | VFPT             | 81.35        | 84.93            | 60.19           |
> | S-CMPT           | **81.84**    | **85.73**        | **60.92**       |
>
> The theoretical reason for this weakness is well-understood.
> A recent theoretical analysis [1] shows:
> **Prepending prompt vectors to a transformer block affects only the subsequent block's attention pattern.**
> It does not modify the attention behavior of the block to which the prompts are attached.
> This implies:
> + The shallow CMPT design (which attaches prompts only to the first Transformer layer) can actively influence only the second transformer block.
> All deeper blocks (layers 3 to the last) are influenced only indirectly.
> Their behavior is modified passively through accumulated hidden-state transformations rather than through direct prompt conditioning.
> + The deep prompt architecture (as used by CMPT-Deep, VFPT, and S-CMPT) attaches prompts to all Transformer layers.
> Thus, it can directly influence the attention patterns of all transformer layers except the first layer.
>
> This difference in controllability explains why shallow CMPT consistently lacks expressiveness compared to deep variants.
>
> [1] Petrov, Aleksandar, Philip Torr, and Adel Bibi. "When Do Prompting and Prefix-Tuning Work? A Theory of Capabilities and Limitations." In The Twelfth International Conference on Learning Representations.
>
>
> ### **Weakness 3: The proposed method is relatively simple. A attention operation is adopted to select prompts from the source model, and a diversity loss is used to prompt source distribution.**
>
> The components of S-CMPT (attention and a diversity regularizer) are simple but well-established.
> We argue this is a primary strength of our work.
>
> The core contribution is not the invention of these modules, but the novel insight into the problem:
> We are the first to empirically prove that "source prompts exhibit redundancy and that strict layer-wise correspondence is suboptimal" in this context.
> Based on this insight, we proposed an elegant and simple "select-then-transfer" framework that solves both problems.
> This "simple" design achieves two significant results:
> 1. It outperforms all more complex baselines, achieving state-of-the-art accuracy.
> 2. It is 2.5x more parameter-efficient than the previous SOTA (VFPT), demonstrating a "superior parameter-performance trade-off".
>
> Our work shows that the key to this problem is selective transfer, not architectural complexity.
>
> ### **Weakness 5: The performance improvement is marginal on most settings. For example, only +0.38%,+2.44%,and +0.18% on VTAB-1k with MoCo V3 pretrained weights.**
>
> We respectfully disagree with the characterization of our improvements as 'marginal.'
>
> 1. Gains over SOTA (VFPT):
> In the highly-optimized VTAB-1k benchmark,
> S-CMPT consistently outperforms SOTA.
> It achieves a +0.65% gain (Supervised ViT),
> a +9.60% gain (MAE ViT),
> and a +1.34% gain (MoCo v3) over VFPT .
> A +0.65% gain over a 2024 SOTA baseline is significant,
> and the ~1-10% gains on self-supervised models are substantial.
>
> 2. Gains over Baselines:
> The reviewer's cited figures (+0.38%, +0.18%) represent the gains over **our re-designed and re-implemented** CMPT,
> not the previous SOTA methods.
>
> 3. The Core Claim (Efficiency-Efficacy):
> The paper's main contribution is not accuracy alone,
> but the parameter-performance trade-off.
> S-CMPT achieves these SOTA results while using only 20 vectors per layer, which is 2.5x fewer prompt vectors than the SOTA VFPT (49.3 vectors) and 3x fewer than VPT (60.9 vectors).
>
> Achieving better performance with 2.5x fewer parameters is the opposite of a marginal improvement; it demonstrates a new, superior trade-off for efficient transfer learning.

---

> > ### Comment · Reviewer_tRV6 · 2025-11-25
> > **Response**
> >
> > Thanks for the response of the authors. Their response has addressed some of my concerns like limited related works and effects of shallow architectures. The authors are encouraged to incorporate them into the paper to better demonstrate their work. Besides, i still have several concerns. First, the proposed method is simple without sufficient design. I agree that the authors are the first to empirically prove that "source prompts exhibit redundancy and that strict layer-wise correspondence is suboptimal" in this context. However, as also indicated by other reviewers, the proposed method is more like a hand-crafted design with intuitive methodology. The authors don't fully convince me about that. Second, the performance is marginal in most cases in my view. This is also pointed by other reviewers, and also reflected in the new results raised by the authors. For example, compared to CMPT-Deep, the proposed method achieve <0.5% performance boost on VTAB-Natural, VTAB-Specialized and VTAB-Structured. This results don't fully convince me. I currently keep my score.

---

> > > ### Author Response · Authors · 2025-11-26
> > >
> > > We appreciate the reviewer's continued feedback.
> > > We are pleased that our responses have addressed concerns regarding related work and the empirical validation of our core hypotheses.
> > >
> > > Regarding your remaining concerns:
> > >
> > > ### **1. The proposed method is simple**
> > >
> > > We agree that our method is simple.
> > > However, we respectfully but fundamentally disagree that this implies it is "without sufficient design" or merely "hand-crafted".
> > > In the field of Parameter-Efficient Transfer Learning (PETL), **simplicity and elegance are hallmarks of sufficient design, not limitations**.
> > >
> > > 1. As you noted, we are the first to empirically prove that "source prompts exhibit redundancy and that strict layer-wise correspondence is suboptimal".
> > > Our modules are the exact necessary solutions to these verified problems.
> > > Adding complexity beyond these necessary components would be over-engineering, which counteracts the efficiency goals of PETL.
> > > 2. Our minimalist philosophy aligns perfectly with the most impactful works in the field,
> > > which prioritize effective mechanisms over architectural complexity:
> > >     + VPT: Simply prepends learnable vectors to inputs.
> > >     + VFPT: Simply transforms vectors to the frequency domain.
> > >
> > > These seminal methods are celebrated because they are simple and elegant.
> > > S-CMPT follows this creed.
> > > We demonstrate that we can outperform these elegant baselines,
> > > not by adding complex architectures,
> > > but by correcting the fundamental assumptions regarding how prompts are selected and transferred.
> > >
> > > Ultimately, we hold the conviction that in the pursuit of parameter efficiency,
> > > simplicity is a virtue, not a shortcoming.
> > > While we deeply respect the reviewer's perspective regarding architectural complexity,
> > > we maintain that S-CMPT's ability to achieve state-of-the-art results through an elegant,
> > > minimalist framework represents a valuable and distinct contribution to the field.
> > >
> > >
> > > ### **2. The performance is marginal in most cases**
> > >
> > > We totally understand and respect your concern.
> > > We hope the following points can solve your concern:
> > >
> > > **2.1 Being marginal does not affect our conclusion**:
> > >
> > > While the absolute margin is small in some settings, the consistency of our improvement proves that S-CMPT is a strictly superior method, not a noisy one.
> > > We analyzed all 57 experimental trials (19 tasks $\times$ 3 backbones) reported in the paper.
> > > S-CMPT outperformed CMPT-Deep in 38 out of 57 trials.
> > > A binomial test yields a p-value of 0.016.
> > > Since $p < 0.05$, we can conclude with statistical confidence that S-CMPT provides a reliable and robust improvement over CMPT-Deep.
> > >
> > > **2.2 Significant magnitude on weak backbones**:
> > >
> > > On strong backbones (Supervised/MoCo), the visual features are already mature (high linear probing performance), creating a ceiling effect.
> > > On MAE where the features are less mature (low linear probing performance),
> > > S-CMPT outperforms CMPT-Deep by +2.44% and the SOTA VFPT by +9.60%.
> > > This demonstrates that when the backbone actually needs external knowledge, S-CMPT's selective mechanism is crucial, whereas the rigid CMPT-Deep struggles.
> > >
> > > **2.3 Strength of CMPT-Deep**:
> > >
> > > It is critical to note that **CMPT-Deep is not a weak, pre-existing baseline**.
> > > The actual prior work is restricted to shallow architectures.
> > > CMPT-Deep is a strong baseline we implemented ourselves to create a rigorous comparison,
> > > which, can be seen as part of our contribution.
> > >
> > > |              | VTAB-Natural | VTAB-Specialized | VTAB-Structured |
> > > |--------------|--------------|------------------|-----------------|
> > > | CMPT-Shallow | 80.55        | 84.12            | 53.47           |
> > > |              |              |                  |                 |
> > > | CMPT-Deep    | 81.48        | 85.55            | 60.44           |
> > > | S-CMPT       | **81.84**    | **85.73**        | **60.92**       |
> > > |              |              |                  |                 |
> > >
> > > As shown above, the jump from prior work (Shallow) to our baseline (Deep) is already significant.
> > > S-CMPT consistently improving upon this strong, self-implemented baseline,
> > > rather than just beating the weaker prior work,
> > > strengthens our conclusions.
> > >
> > > **Summary**:
> > >
> > > We achieve statistically significant consistency,
> > > massive gains on weaker backbones,
> > > and superior parameter efficiency against a strong existing baseline VFPT.
> > > We believe this constitutes a robust and non-marginal contribution.

---

### Official Review · Reviewer_rnza · 2025-11-01

**Soundness:** 3
**Presentation:** 3
**Contribution:** 3
**Rating:** 6
**Confidence:** 4

**Summary:**

The paper studies Cross-Modality Prompt Transfer (CMPT) in deep architectures and proposes S-
CMPT, a select-then-transfer pipeline. For each target ViT layer, an attention-based selector picks
vectors from a global pool of source prompt vectors (flattened across all source layers), and a single
linear projector adapts them to the target layer. During training, only selectors and projectors are
optimized; at inference they are discarded, and only the selected prompts remain. On VTAB-1k with
ViT-B/16, S-CMPT achieves strong performance compared to VPT, VFPT, and other parameter-
efficient tuning baselines on both supervised and self-supervised (MAE, MoCo) ViTs. A pilot study
shows that random cross-layer selection beats CMPT in 34 / 36 runs, suggesting redundancy in
source prompts and the limits of rigid layer correspondence.

**Strengths:**

- Clear motivation. The cross-layer pilot experiment convincingly demonstrates redundancy and cross-
layer compatibility, motivating a global-pool selection strategy.
- Simple and principled. The method uses a lightweight attention selector with diversity regularization to
reduce redundant picks; only selector + projector are trained, while inference keeps just the selected
prompts.
- Consistent gains. The approach performs competitively or better than VPT/VFPT across all VTAB-1k
groups and pretraining types, with fewer tunable parameters overall.

**Weaknesses:**

- Lack of Np sensitivity study. The method fixes Np = 20 without justification or sweeps (e.g., 4–64)
showing accuracy/memory/throughput trade-offs across task groups and pretraining types.
- Incomplete efficiency analysis. Although inference removes selectors/projectors, the paper omits end-
to-end wall-clock, memory, FLOPs, and throughput comparisons—especially the selector’s attention-
based cost and its scaling with Nl, Np, dh.
- Unclear robustness and negative-transfer boundary. While selection-scope/diversity ablations and
visualizations exist, there are no failure cases, temperature τ or entropy diagnostics, or confidence-
thresholded variants to handle worst-case selections.

**Questions:**

- Could you provide Np sweeps (e.g., 4–64) and accuracy curves per VTAB group？ Is there a stable
operating range?
- It would strengthen the paper to report end-to-end training/inference metrics versus VPT/VFPT/LoRA:
wall-clock time, peak memory, FLOPs, and throughput. Break down the selector cost.
- Do you observe harmful selections on “distant” tasks? Can τ-adaptation or confidence-thresholded
selection reduce worst-case drops? Please include failure cases and correlations between selection
entropy / max-attention and accuracy.
- The selector uses full-pool attention, which may be costly. Could lighter options—like sparse routing or
clustering-based selection—achieve similar adaptivity with lower compute?

---

> ### Author Response · Authors · 2025-11-19
> **Response to question 1 & 2**
>
> ### **1. Could you provide Np sweeps (e.g., 4–64) and accuracy curves per VTAB group？ Is there a stable operating range?**
>
> We thank the reviewer for this important question.
> Our choice of Np=20 was intentionally fixed to demonstrate simplicity,
> which is a core contribution of S-CMPT. S-CMPT "eliminates the need for exhaustive prompt-length tuning".
> We show that a "fixed length of 20 is sufficient to achieve strong performance".
> This avoids the extra computational overhead of finding an optimal prompt length for every task,
> which is a common burden in other methods.
>
> However, for exploration,
> we have conducted a Np sweep on the grid [5, 10, 20, 50, 100],
> which is the same grid adopted in the seminal work Visual Prompt Tuning [1] (which is also why we choose this grid).
> The results on three representative tasks (Cifar, Resisc45, and Kitti) are as follows:
>
> | Np       | 5          | 10         | 20             | 50             | 100          |
> |----------|------------|------------|----------------|----------------|--------------|
> | Cifar    | 81.28±0.33 | 82.21±0.25 | 82.70±0.25     | **82.78±0.34** | 52.32±29.48  |
> | Resisc45 | 84.59±0.48 | 85.10±0.61 | **85.72±0.24** | 85.26±0.68     | 85.02±0.81   |
> | Kitti    | 79.00±0.49 | 79.28±1.35 | **80.31±0.86** | 79.37±1.42     | 78.72±1.18   |
>
> We can see that:
> 1. Performance steadily improves from Np=5 to Np=20,
> which can be regarded as a stable operating range (as the performance scales with prompt length).
> This empirically validates our original choice.
> 2. Increasing the length further to Np=50 or 100 provides no significant benefit and even introduces instability.
> This is likely caused by transferring too much prompt vectors,
> which could introduce noise and make the optimization task more difficult.
>
> [1] Jia, Menglin, Luming Tang, Bor-Chun Chen, Claire Cardie, Serge Belongie, Bharath Hariharan, and Ser-Nam Lim. "Visual prompt tuning." In European conference on computer vision, pp. 709-727. Cham: Springer Nature Switzerland, 2022.
>
>
> ### **2. It would strengthen the paper to report end-to-end training/inference metrics versus VPT/VFPT/LoRA: wall-clock time, peak memory, FLOPs, and throughput. Break down the selector cost.**
>
> We thank the reviewer for this crucial point, which allows us to clarify our efficiency claims.
> Our paper's "Efficiency & Efficacy" claim is focused on the final parameter-performance trade-off,
> which is a central goal of Parameter-Efficient Transfer Learning (PETL).
> However, to thoroughly address the reviewer's valid concern about training cost,
> we have benchmarked S-CMPT against the baselines.
>
> The results show that the overhead from our selector and projector is marginal and does not create a computational bottleneck:
>
> |        | GPU Time (Train) | GPU Memory (Train) | FLOPs (Train) |
> |--------|------------------|--------------------|---------------|
> | S-CMPT | 429.840 ms       | 5.57 GB            | 26.90 GFLOPs  |
> | VPT    | 428.203 ms       | 5.51 GB            | 26.62 GFLOPs  |
> | VFPT   | 433.391 ms       | 5.51 GB            | 26.62 GFLOPs  |
>
> _All benchmarks run with a batch size of 64 and Np=20.
> Time is averaged over 100 steps.
> FLOPs are for batch size 1.
> The script is `efficiency_comparison.py` in the newly updated .zip file_
>
> This data leads to two key conclusions:
> 1. Training Overhead is Marginal:
> The reviewer correctly identified that our attention-based selector and projector add computational cost (but is marginal).
> This is reflected in the minor increases in training memory (+0.06 GB) and FLOPs (+0.28G FLOPs) over the baselines.
> 2. S-CMPT is Faster than the SOTA Baseline:
> Importantly, S-CMPT's training time is faster than the SOTA baseline VFPT
> (which incurs costs for Fourier transforms) and is statistically negligible (+1.6 ms) compared to the simpler VPT.
>
> This confirms our design choice:
> we accept a minor, one-time training overhead to enable our selection mechanism.
> This trade-off unlocks S-CMPT's primary efficiency gains at inference time.
>
> As the reviewer notes, our selector and projector "are discarded" after training.
> S-CMPT's inference cost is therefore determined only by its final, compact prompt length (Np=20).
> This makes our method ~2.5x more parameter-efficient (20 vs. 49.3 vectors) and correspondingly faster at inference (fewer vectors to process) than VFPT,
> all while achieving superior accuracy.
>
> The computational efficiency comparison has been added to the discussion section in our supplementary material.

---

> ### Author Response · Authors · 2025-11-19
> **Response to question 3**
>
> ### **3. Do you observe harmful selections on “distant” tasks? Can τ-adaptation or confidence-thresholded selection reduce worst-case drops? Please include failure cases and correlations between selection entropy / max-attention and accuracy.**
>
> **Harmful selections**
>
> Unfortunately, we did not observe harmful selections.
> However, we can manually create harmful selections simply by setting the diversity regularization weight lambda to -0.01.
> This actively maximizes cosine similarity, forcing the model to select repetitive, redundant vectors.
>
> We then analyzed the correlations on CIFAR
> (we choose CIFAR for its hyperparameter stability:
> no large-scale hyperparameter tuning is needed.
> _Code details: the `prompt_stats_analysis` function in `train_utils.py`._).
> The metrics are:
> + Selection Entropy: The avg. entropy of the aggregated softmax attention. Lower = more repetitive selection.
> + Max Attention: The avg. max value of the aggregated softmax attention. Higher = more repetitive selection.
> + \# Vectors (A > 0.5): The avg. count of source vectors with significant attention (>0.5). Lower = more repetitive selection.
>
> | Seed | Np | lambda | Acc   | Selection Entropy | Max Attention | # Vectors (A > 0.5) |
> |------|----|--------|-------|-------------------|---------------|---------------------|
> | 42   | 20 | -0.01  | 81.52 | 2.4800            | 10.5205       | 6.6667              |
> | 42   | 20 | 0      | 81.76 | 4.2943            | 2.3368        | 16.6667             |
> | 42   | 20 | 0.01   | 82.86 | 4.1741            | 2.2496        | 18.5833             |
> |      |    |        |       |                   |               |                     |
> | 44   | 20 | -0.01  | 81.83 | 3.7249            | 9.2198        | 3.2500              |
> | 44   | 20 | 0      | 82.34 | 4.1644            | 3.1758        | 14.3333             |
> | 44   | 20 | 0.01   | 82.69 | 4.1663            | 1.8331        | 18.4167             |
> |      |    |        |       |                   |               |                     |
> | 100  | 20 | -0.01  | 80.28 | 2.7226            | 10.7365       | 4.5833              |
> | 100  | 20 | 0      | 82.03 | 4.2765            | 2.7527        | 15.9167             |
> | 100  | 20 | 0.01   | 82.06 | 4.2264            | 1.7527        | 18.9167             |
>
> The data clearly supports the reviewer's intuition:
> 1. The lambda = -0.01 case consistently creates harmful selections, validated by all three metrics:
> it has critically low entropy, high max-attention, and attends to far fewer source vectors.
> 2. This "selection collapse" directly correlates with worse performance:
> On all seeds, the lambda = -0.01 setting had the lowest accuracy.
> This confirms our hypothesis that selection redundancy is detrimental.
>
> **τ-adaptation or confidence-thresholded selection**
>
> The reviewer's next question (can these techniques reduce such "worst-case drops") is an excellent one.
> We implemented both:
> + τ-adaptation: We added a per-layer learnable temperature scalar to our selector.
> + Confidence-thresholded selection:
> As a hard threshold is non-differentiable, we used Gumbel-Softmax [1].
> This is a fully differentiable,
> "hard" selection mechanism that outputs a one-hot vector,
> perfectly implementing the idea of "thresholding" to select only the single most important vector per query.
> + Code: class `PromptSelectorNew` in `./model/vpt.py`.
>
> We then tested this new, more robust selector (S-CMPT+τ+Gumbel) in the same hostile environment (lambda = -0.01):
>
> | Seed | Np | lambda | Acc   | Selection Entropy | Max Attention | # Vectors (A > 0.5) |
> |------|----|--------|-------|-------------------|---------------|---------------------|
> | 42   | 20 | -0.01  | 81.80 | 2.9653            | 6.1667        | 11.0833             |
> | 44   | 20 | -0.01  | 81.58 | 2.5893            | 8.0833        | 9.3333              |
> | 100  | 20 | -0.01  | 81.49 | 2.3715            | 8.7500        | 8.7500              |
>
> The results are very promising.
> In 2 out of 3 seeds (42 and 100),
> the new method successfully fought the 'harmful' pressure from lambda=-0.01.
> It achieved higher accuracy (e.g., 81.49 vs 80.28)
> because it was able to find less repetitive selections (e.g., lower max attention, more vectors).
>
> Even on Seed 44,
> where the new method underperformed,
> the metrics still support our hypothesis:
> it produced more repetitive selections (lower entropy) and thus lower accuracy,
> further validating the correlation.
>
> This confirms the reviewer's suggestions are highly effective at mitigating "worst-case drops."
> This is a very promising direction for improving S-CMPT's robustness.
> We thank the reviewer deeply for this wonderful suggestion.
> Given the limited rebuttal period,
> we will have to leave a large-scale exploration to our future work.
>
> [1] Jang, Eric, Shixiang Gu, and Ben Poole. "Categorical Reparameterization with Gumbel-Softmax." In International Conference on Learning Representations. 2017.

---

> ### Author Response · Authors · 2025-11-19
> **Response to question 4**
>
> ### **4. The selector uses full-pool attention, which may be costly. Could lighter options—like sparse routing or clustering-based selection—achieve similar adaptivity with lower compute?**
>
> We thank the reviewer for this excellent suggestion for architectural exploration.
>
> First,
> regarding the concern of cost,
> our new efficiency analysis (in response to Q2) shows our full-pool attention mechanism is not a practical bottleneck.
> Our benchmarks show S-CMPT's training time is comparable to VPT (+1.6 ms) and is,
> in fact, faster than the current state-of-the-art baseline, VFPT.
> This demonstrates that our simple attention module is highly practical.
>
> Regarding the specific alternatives:
> + Clustering-based Selection:
> We believe this approach, while interesting, introduces its own complexities.
> A standard clustering (e.g., k-means) operates statically only on the source vectors.
> To achieve the adaptivity of S-CMPT (where the selection is a function of the target data and model layer)
> would require a more complex, learnable clustering design.
> Furthermore, iterative clustering algorithms could be less efficient than our single, parallelizable matrix multiplication.
> + Sparse Routing:
> We agree this is a highly promising direction.
> Our paper's primary contribution was to prove the principle that a "select-then-transfer" paradigm is necessary and that a full,
> layer-agnostic selection is superior to a rigid, local one.
> Having now validated this principle,
> exploring sparse routing mechanisms is the logical "next step" to further optimize the training-time computation.
>
> This is an excellent avenue for future research, we sincerely thank the reviewer for pointing this out!

---

### Author Response · Authors · 2025-11-19
**Updates on the Revision**

Dear area chair and reviewers,

We sincerely thank you for your time and the dedicated effort put into reviewing our work.
Your insightful comments have prompted us to conduct deeper analyses and additional experiments,
which we believe have significantly strengthened the quality and rigor of our submission.
Based on your feedback, we have updated our manuscript and supplementary material with the following key improvements:
1. We have expanded the related work section to discuss more contemporary literature.
2. We have added a theoretical basis for our proposed method to our supplementary material.
3. We have added a discussion section in our supplementary material, focusing on the training efficiency and limitations of S-CMPT.

We once again express our deepest gratitude for your guidance in refining this work!

Best Wishes,

Authors

---

### Meta-Review · Area_Chair_tgHo · 2025-12-28

**Summary:**

The paper proposes Selective Cross-Modality Prompt Transfer (S-CMPT), which selectively transfers a subset of prompt vectors from a source modality (e.g., text) to address issues like redundancy and incompatibility in cross-modality prompt transfer for data-scarce tasks. Some reviewer concerns were addressed such as sensitivity to prompt length, efficiency benchmarks, harmful selections, and selector costs. While the work demonstrates some empirical improvements and achieves greater efficiency, the gains appear marginal compared to strong baselines. Moreover, it does not address concerns about a lack of innovation and limited theoretical depth.

Besides, motivation could be stronger by explicitly comparing with dual-encoder models like CLIP, especially given their dominance in cross-modal transfer; without this, contributions appear incremental. Thus, I recommend rejecting this paper. Thank you for flagging Pynv. This feedback has been taken into account in the decision.

**Reviewer Concerns:**

Concerns such as novelty and marginal gained performance are still unaddressed.

**Reviewer Scores:**

No reviewer would change the score.

---

### Decision · Program_Chairs · 2026-01-26

Reject